# OW-VAP: Visual Attribute Parsing for Open World Object Detection

Xing Xi [1 2]   Xing Fu [2]   Weiqiang Wang [2]   Ronghua Luo [1]

## Abstract

Open World Object Detection (OWOD) requires the detector to continuously identify and learn new categories. Existing methods rely on the large language model (LLM) to describe the visual attributes of known categories and use these attributes to mark potential objects. The performance of such methods is influenced by the accuracy of LLM descriptions, and selecting appropriate attributes during incremental learning remains a challenge. In this paper, we propose a novel OWOD framework, termed OW-VAP, which operates independently of LLM and requires only minimal object descriptions to detect unknown objects. Specifically, we propose a Visual Attribute Parser (VAP) that parses the attributes of visual regions and assesses object potential based on the similarity between these attributes and the object descriptions. To enable the VAP to recognize objects in unlabeled areas, we exploit potential objects within background regions. Finally, we propose Probabilistic Soft Label Assignment (PSLA) to prevent optimization conflicts from misidentifying background as foreground. Comparative results on the OWOD benchmark demonstrate that our approach surpasses existing state-of-the-art methods with a +13 improvement in U-Recall and a +8 increase in U-AP for unknown detection capabilities. Furthermore, OW-VAP approaches the unknown recall upper limit of the detector.

## 1. Introduction

Object detection is a fundamental task in the field of computer vision, aiming to classify and localize objects within an image (Tian et al., 2019; Lin et al., 2017b). Traditional object detectors (Li et al., 2020b; Liu et al., 2016), often

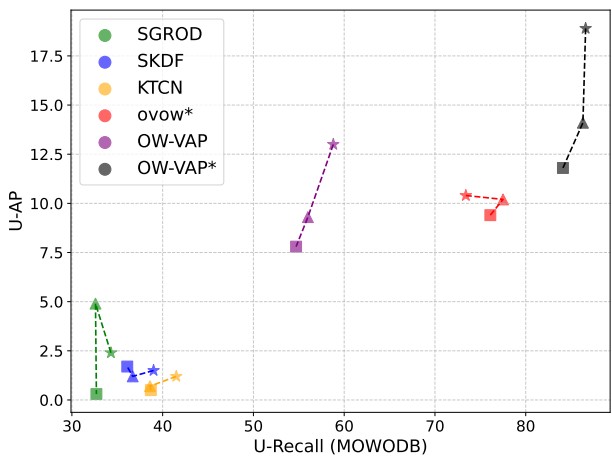

Figure 1. **Comparison Curves for U-AP and U-Recall**. All metrics are evaluated on MOWODB benchmark. ★, △ and □ denote the performance in Task 1, Task 2, and Task 3, respectively. ∗ indicates the performance after removing duplicate images. Our OW-VAP significantly outperforms other models.

referred to as closed-set detectors, are trained on a fixed set of categories and, therefore, are only capable of recognizing these predefined categories. However, in real-world scenarios, annotating all possible categories is impractical. To address this limitation, Open World Object Detection (OWOD) has been proposed (Joseph et al., 2021). The detection of new categories in OWOD is divided into two steps. Initially, the detector is trained on a dataset with a fixed set of categories and is required to detect objects that were not labeled in the training set during the inference phase. Subsequently, these objects are selected and introduced in subsequent incremental learning processes. The detector is then fine-tuned based on existing knowledge, thereby gaining the ability to detect these new categories.

Existing work can be broadly classified into two types. The first assumes that potential objects share certain similarities with known categories. Detectors demonstrate anomalies when they label objects similar to known categories as negative samples. OW-DETR (Gupta et al., 2022) assesses feature maps corresponding to negative sample regions, selecting the top K samples with the highest scores as potential objects. The second utilizes the knowledge of visual foun-

[1]School of Computer Science & Engineering, South China University of Technology, Guangzhou, China [2]Ant Group, Hangzhou, China. Correspondence to: Ronghua Luo <rhluo@scut.edu.cn>, Xing Fu <zicai.fx@antgroup.com>.

*Proceedings of the $42^{nd}$ International Conference on Machine Learning*, Vancouver, Canada. PMLR 267, 2025. Copyright 2025 by the author(s).

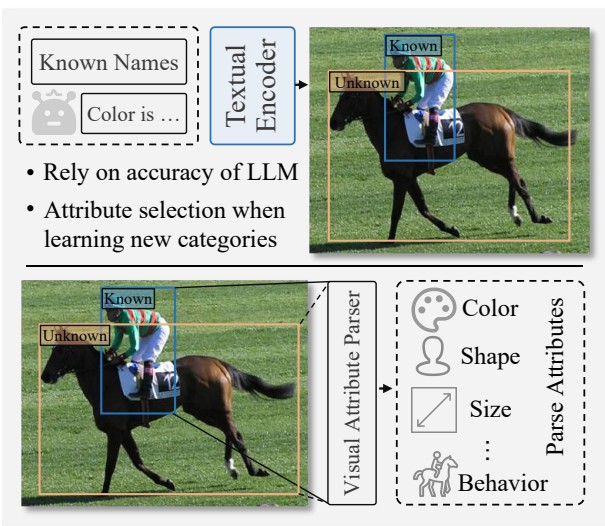

*Figure 2.* **Unknown Object Detection Illustration**. The top section illustrates the previous OVD-based OWOD framework, which relies on LLM for object descriptions, thereby being constrained by the accuracy of LLM. During incremental learning, balancing the contribution of each attribute is challenging. The bottom section depicts our proposed OW-VAP, which learns to parse attributes directly from regions without relying on guidance from LLM.

dation models for supervision. KTCN (Xi et al., 2024) uses the Segment Anything Model (SAM) (Kirillov et al., 2023) to generate candidate boxes for all objects in the image and selects candidates that do not overlap with labeled objects as pseudo labels. Recently, leveraging the generalization capabilities of Open Vocabulary object Detection (OVD) to detect unknown objects has become the mainstream, achieving significant recall advantages.

As depicted in Figure 2 (top), these detectors assume that unknown objects share certain attributes with known categories, such as the color blue in the appearance of the object. During inference, objects exhibiting similarities in these attributes are labeled as unknown category. However, these detectors have two main drawbacks. Firstly, all predictions are based on the attributes provided by Large Language Model (LLM). Consequently, their performance is highly susceptible to the accuracy of attribute descriptions by LLM. Secondly, in subsequent incremental learning, it is challenging to quantify the contribution of each attribute to the unknown class, leading to difficulties in attribute selection. Based on these limitations, we propose a novel detection framework, termed OW-VAP, in this paper.

As shown in Figure 2 (bottom), the key to OW-VAP lies in enabling the detector to learn to parse the attributes of objects. Specifically, we utilize a set of vague sentences to describe an object, focusing on attributes such as shape, scale, and color. Subsequently, we train a Visual Attribute

Parser (VAP) to parse the attributes corresponding to visual regions. Since VAP is trained on known classes (Fang et al., 2023), there exists a label bias towards the attributes of known objects. Therefore, during VAP training, we mine potential objects in background samples as pseudo labels. However, these pseudo labels do not always correctly encompass objects. To prevent optimization conflicts arising from pseudo labels, we assign soft labels to pseudo labels based on loss distribution probabilities. Finally, the trained VAP can parse the attributes corresponding to a given object region to estimate whether it contains an object, independent of the fine-grained attributes provided by LLM.

We evaluate OW-VAP on standard OWOD benchmarks, MOWODB and SOWODB, which are composed of a mixture of VOC (Everingham et al., 2010) and COCO (Lin et al., 2014) datasets. In Figure 1, we present the curves of U-Recall and U-AP on MOWODB benchmark, which are the primary metrics of interest for OWOD. OW-VAP outperforms the previous state-of-the-art (SOTA) methods, by a margin of 13+ U-Recall. Furthermore, on stricter evaluation metrics, we achieve an 8+ U-AP performance advantage. Notably, OW-VAP has reached or even surpassed the upper limit of OVD's generalization capability in terms of recall. In Task 2, we surpass the recall upper limit of OVD using all class names by 0.3 U-Recall, achieving 56.3 U-Recall. We summarize our contributions as follows:

- We propose OW-VAP, a novel detection framework for OWOD that does not rely on guidance from LLM.

- We propose the visual attribute parser (VAP) to parse the visual attributes corresponding to the current region, assessing the likelihood of containing objects.

- We propose the probabilistic soft label assignment (PSLA) to mitigate optimization conflicts arising from background noise.

- We validate the effectiveness of OW-VAP on the standard OWOD benchmark. OW-VAP surpasses the state-of-the-art (SOTA) methods with an advantage of over 13 U-Recall and 8 U-AP in unknown detection. Furthermore, OW-VAP exceeds the generalization upper limit of OVD in Task 2 of MOWODB.

## 2. Related Work

### 2.1. Object Detection

Object detection (OD), a fundamental task in computer vision, aims to detect and localize objects of interest within images. Early detection methods employed a two-stage process, dividing the detection task into candidate proposal and refinement stages (Girshick, 2015; Ren et al., 2015). The former provides coarse, class-agnostic bounding boxes

for objects, while the latter refines these boxes and performs classification. However, due to the slower detection speed of such methods, single-stage detection approaches have been proposed (Wang et al., 2021; Chen et al., 2021). Unlike their two-stage counterparts, single-stage methods directly perform dense predictions on the downsampled feature maps. Typical representatives include SSD (Liu et al., 2016) and the YOLO series (Ge et al., 2021; Wang et al., 2023a), which have become the preferred choice in the industry (Zhang et al., 2022; Wu et al., 2021). In recent years, the sequence attention mechanism of transformers has been proven effective in the field of natural language processing. Consequently, DETR (Carion et al., 2020) transformed the detection task into set prediction, successfully introducing this technology to OD. In addition, some studies have focused on addressing issues arising during OD training and inference, including multi-scale feature fusion (Lin et al., 2017a; Tan et al., 2020), non-maximum suppression (Sun et al., 2021; Wang et al., 2020), enhancing Transformer convergence speed (Roh et al., 2022; Pu et al., 2023), training data imbalance(Lin et al., 2017b), and bounding box ambiguity (Li et al., 2020b;a). Although these methods have played a significant role in advancing the development and industrial application of object detection technology, they generally follow a closed-set setting, recognizing only the categories annotated in the training data.

## 2.2. Open Vocabulary Object Detection

Open Vocabulary Object Detection (OVD), an emerging field, was introduced by OV-RCNN (Zareian et al., 2020). OVD redefines the classification process of the detection head as a matching task between class names and regions. he advantage of this matching process is that the number of matches can dynamically vary. Thus, OVD detectors can detect an arbitrary number of classes (Wang et al., 2023d; Saito et al., 2022). Furthermore, benefiting from large-scale pre-training model. such as CLIP (Radford et al., 2021), OVD can demonstrate zero-shot detection advantages for completely unseen classes (Fang et al., 2024; Jin et al., 2024). Current OVD research focuses on aligning regions with text (Wu et al., 2023a; Yao et al., 2023), knowledge distillation (Wang et al., 2023b; Gu et al., 2022), learning region prompts (Feng et al., 2022; Wu et al., 2023b), and large-scale pre-training (Yao et al., 2022; Li et al., 2022). These approaches enable OVD to exhibit strong visual encoding capabilities, sometimes even surpassing fully supervised closed-set detectors in certain scenarios (Wang et al., 2023d). However, OVD still faces several limitations in practical applications. The main issue is that, despite its theoretical ability to detect any category, OVD still relies on a predefined set of categories for text matching in practice. This reliance restricts its capacity to adapt to and handle new categories in dynamically changing environments, limiting its ability to surpass the constraints of closed-set detectors.

## 2.3. Open World Object Detection

Open World Object Detection (OWOD), distinct from OVD, employs a two-stage process for recognizing new objects (Joseph et al., 2021). Initially, the detector is trained on a closed set of classes. During inference, it identifies potential objects of interest. These objects are then selected by annotators and subjected to incremental learning for fine-tuning. Consequently, predicting potential objects during inference is crucial. Prior to the introduction of foundation models, researchers focused on generating pseudo-labels for unknown classes (Zohar et al., 2023b; Zhao et al., 2022). ORE leverages the class-agnostic capabilities of the Region Proposal Network (RPN) to mark high-scoring background samples as potential objects. OW-DETR scores background regions and selects high-scoring backgrounds as pseudo-labels. CAT, UC-OWOD, and RandBox (Ma et al., 2023a; Wu et al., 2022; Wang et al., 2023c) utilize selective search to provide candidates. The introduction of foundation models significantly enhances the capabilities of OWOD detectors. KTCN and SGROD (Xi et al., 2024; He et al., 2024) distill knowledge from SAM (Kirillov et al., 2023) to generate pseudo-labels, while SKDF and ovow (Ma et al., 2024; Li et al., 2024) leverage the generalization capabilities of OVD detectors. These methods have achieved significant performance breakthroughs. In this paper, we propose OW-VAP, which enables detectors to learn to parse the attributes of visual regions without relying on guidance from LLM.

## 3. Method

### 3.1. Overall

The architecture of OW-VAP, as illustrated in Figure 3, builds upon the YOLO-World detector (Cheng et al., 2024). Compared to the standard OVD detector, our model incorporates two additional components: coarse textual sentences and Visual Attribute Parser (VAP). The textual sentences provide generic attributes that describe objects, while the VAP learns to parse these attributes from visual embeddings (Section 3.2 and Section 3.3). During the inference phase for known classes, we employ the OVD inference process, which classifies objects based on the cosine similarity between visual and class name embeddings. For the detection of unknown classes, we utilize the VAP and attribute texts to estimate the likelihood that a region contains an object (Section 3.4). Finally, we apply standard OD post-processing, including non-maximum suppression (NMS) and ranking, to bounding boxes for both known and unknown classes to yield the final predictions. The problem definition of OWOD and pseudocode for the proposed components (VAP, PSLA) are shown in Appendix B and Appendix H, respectively.

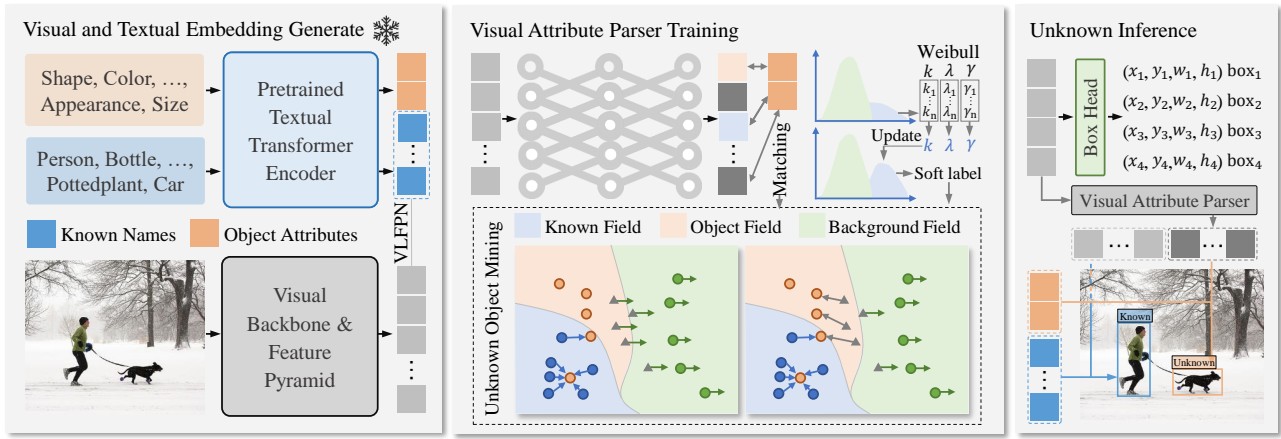

Figure 3. **Training and Inference Pipeline of OW-VAP**. The OW-VAP framework is built on the standard OVD detector, YOLO-World (Cheng et al., 2024). During training, known class names and vague textual attributes are encoded into text embeddings via a text encoder, while images are converted into visual embeddings through a visual encoder (left, Visual Backbone & Feature Pyramid (Lin et al., 2017a)). The Visual Attribute Parser (VAP) learns to extract the corresponding textual attributes from these visual embeddings (middle). The trained VAP is utilized to estimate the likelihood of an object being present in a given region. By combining the VAP's estimates with the similarity of vague text, unknown objects are identified (right).

## 3.2. Visual Attribute Parser

After completing the training phase, we expect the Visual Attribute Parser (VAP) to effectively interpret the attributes associated with the visual embeddings. To achieve this, we predefined a set of object descriptions that differ from the detailed sentences typically generated by LLM. These descriptions focus on determining whether certain attributes are prominent, such as $The\ object\ has\ a\ noticeable\ color$ (refer to the details in Appendix A). Then, the text encoder processes these sentences and known class names to generate attribute and category embedding:

$$E_{att} = \text{Enc}_t(Att),\ E_c = \text{Enc}_t(Name), \quad (1)$$

where $\text{Enc}_t$ denotes the text encoder, specifically the pretrained CLIP text encoder (Radford et al., 2021). $E_{att}$ represents the attribute embedding, while $E_c$ denotes the class embedding. $Att$ and $Name$ represent the attribute sentences and class names, respectively. Subsequently, we employ the vision encoder ($\text{Enc}_v$) to obtain the visual embeddings $E_{vis} = \text{Enc}_v(I)$, where $I$ denotes the input image. The VAP, which is composed of a simple multilayer perceptron, learns to extract attributes from $E_{vis}$. To ascertain the positive samples during training, we rely on the matching process of known classes [1]. Specifically, we utilize the matching results of known classes to filter both positive and negative samples:

$$S = \text{argmin}_\pi \triangle(E_c, E_{vis}, Y_I), \quad (2)$$

---

[1]For clarity, we omit the processes of the VLFPN (Cheng et al., 2024), classification and regression head.

where $Y_I$ denotes the annotation of the current image, encompassing known class labels and bounding boxes. $\triangle$ denotes the matching score of the current region to known classes, encompassing both classification and localization metrics, such as CIOU (Zheng et al., 2020). The matching method ($\pi$) is employed to execute selection based on matching scores, such as the TAL (Feng et al., 2021). $S \in [0, 1]$ represents the value assigned to each sample post-matching, where 0 indicates negative sample.

Subsequently, we label the visual embeddings corresponding to the positive samples as positive samples for VAP after parsing. However, as depicted in Figure 3 (middle), there exist some potential objects in the background. If only known labels are used for matching, it causes VAP to learn extraction representations solely of known classes, leading to bias. Consequently, we label some pseudo-labels from the background. We establish three conditions for annotating pseudo-labels. Due to the generalization capability of OVD, predictions with high similarity are more likely to be positive sample. The similarity to textual attributes must be greater than or equal to mean similarity of positive samples:

$$\text{Cond 1:} \quad \max_{|Att|}\text{Sim}(E_{vis}, E_{att}) \geq$$
$$\text{mean}_{|E_{vis}^+|}\max_{|Att|}\text{Sim}(E_{vis}^+, E_{att}), \quad (3)$$

where $\text{Sim}(\cdot)$ denotes cosine similarity, while $\max_{|\cdot|}$ and $\text{mean}_{|\cdot|}$ refer to the maximum and mean values, respectively. $E_{vis}^+$ represents the positive sample visual embeddings selected through the matching method. When the similarity of all attributes is uniformly low, there is a higher likelihood that the samples selected by Cond 1 contain background

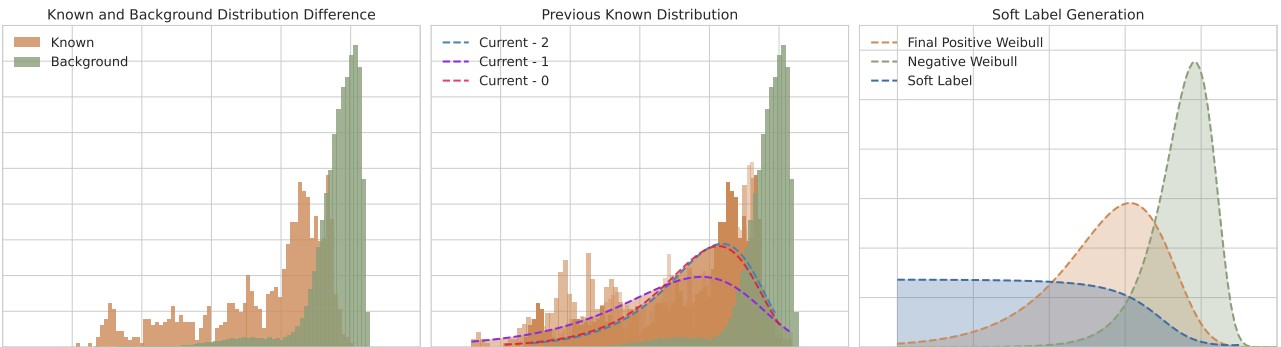

*Figure 4.* **Probabilistic Soft Label Assignment Pipeline**. The left illustrates the loss distributions, where Known and Background represent the loss distributions for the positive and the background samples as generated by the assignment method, respectively. Current-1 and Current-2 represent the fitting results from one and two epochs prior, respectively, while Current-0 denotes the current result. The right represents the generation of soft labels, where the Final Positive Weibull and Negative Weibull denote the final positive sample distribution and the current negative sample distribution, respectively. Soft Label refers to the distribution of the generated pseudo-labels.

noise. Consequently, Cond 2 employs a lower threshold to further refine the dataset by filtering out these samples:

$$\text{Cond 2:} \quad \max_{|Att|} \text{Sim}(E_{vis}, E_{att}) \geq \alpha, \quad (4)$$

where $\alpha$ is a hyperparameter that controls the minimum threshold of similarity. Finally, consistent with the previous methods ([Joseph et al., 2021](#); [Gupta et al., 2022](#)), we select only the top k samples with the highest similarity from the background as candidates:

$$\text{Cond 3:} \quad \text{Top}_\delta \left( \max_{|Att|} \text{Sim}(E_{vis}, E_{att}) \right), \quad (5)$$

where $\delta$ is a hyperparameter used to control the number of selections. We consider a sample as a potential target and assign it a pseudo-label only if it satisfies all three conditions simultaneously, $S_{vap} = 1(\text{Cond 1} \wedge \text{Cond 2} \wedge \text{Cond 3})$, where $1(\cdot)$ is the indicator function, which equals 1 if the condition is satisfied and 0 otherwise.

### 3.3. Probabilistic Soft Label Assignment

To identify high-quality candidates, we employ three conditions to filter out background. However, the background still infiltrates pseudo-labels, leading to optimization conflicts. To address this, inspired by recent work ([Fang et al., 2023](#)), we propose Probabilistic Soft Label Assignment (PSLA), which estimates the probability of pseudo-labels being true positive samples based on their loss distributions.

The training of VAP minimizes the distance to the attributes in the embedding space. When all attributes are treated as positive samples and the closest attributes are pulled closer, the loss exhibits an inconsistent distribution. As shown in Figure 4 (left), the loss in the labeled sample is lower compared to the background region, distributed to the left of the background sample. Background samples also exhibit a consistent pattern, with higher losses distributed to the

right. Therefore, we exploit these distribution differences to balance the optimization conflicts.

$$\text{Wb}(x|\lambda, k, \theta) = k(x-\theta)^{k-1}\lambda^{-k}e^{-(x-\theta)^k\lambda^{-k}}, \quad (6)$$

where $\lambda$, $k$, and $\theta$ represent the scale, shape, and location parameters, respectively. To determine the distribution parameters of positive and negative samples, we employ maximum likelihood estimation. However, the distribution of positive samples is sparser compared to that of negative samples, resulting in greater fluctuations in the loss distribution. To address this, we utilize the moving weighted average to mitigate these fluctuations. As shown in Figure 4 (middle), we retain the historical distribution of positive samples, extract the corresponding distribution parameters, and maintain a queue of these parameters. After modeling in the current epoch, the corresponding parameters are added to the parameter queue. Subsequently, we calculate the weighted average in the queue to generate the final distribution:

$$\lambda^+ = \sum_{i=1}^n \lambda_i^+(2i-1)n^{-2}, \quad (7)$$

where $n$ denotes the length of the queue. $\lambda_i^+$ represents the $i$-th scale parameter within the queue of positive sample parameters. The coefficients used here are derived from an arithmetic sequence designed to emphasize the contribution of the most recently added distribution to the queue. The first term and common difference of this sequence are denoted by $n^{-2}$ and $2n^{-2}$, respectively. The generated soft labels (Figure 4, right) are:

$$\text{W}(x) = \text{Wb}^+(x)\left(\text{Wb}^+(x) + \text{Wb}^-(x)\right)^{-1}, \quad (8)$$

where $Wb^+(\cdot)$ and $Wb^-(\cdot)$ represent the positive sample distribution (as shown in Equation (7)) and the negative sample distribution, respectively. Ultimately, we employ the

*Table 1.* **Performance Comparison on the OWOD Standard Evaluation Benchmark.** The top section shows MOWODB, while the bottom section shows SOWODB. Previously Known and Current Known represent the categories introduced in previous and current tasks, respectively. Both indicates average performance across all previously seen categories. * denotes evaluation after deduplication of the test set. Zero-Shot (GT) represents zero-shot testing with unknown class names to evaluate the OVD detector's generalization upper bound.

| Task IDs (→) | Task 1 | | Task 2 | | | | Task 3 | | | | Task 4 | | |
|---|---|---|---|---|---|---|---|---|---|---|---|---|---|
| | U-Recall | mAP (↑) | U-Recall | mAP(↑) | | | U-Recall | mAP(↑) | | | mAP(↑) | | |
| Method | (↑) | Current Known | (↑) | Previously Known | Current Known | Both | (↑) | Previously Known | Current Known | Both | Previously Known | Current Known | Both |
| ORE_EBUI (Joseph et al., 2021) | 4.9 | 56.0 | 2.9 | 52.7 | 26.0 | 39.4 | 3.9 | 38.2 | 12.7 | 29.7 | 29.6 | 12.4 | 25.3 |
| OW-DETR (Gupta et al., 2022) | 7.5 | 59.2 | 6.2 | 53.6 | 33.5 | 42.9 | 5.7 | 38.3 | 15.8 | 30.8 | 31.4 | 17.1 | 27.8 |
| ALLOW (Ma et al., 2023b) | 13.6 | 59.3 | 10.0 | 53.2 | 33.5 | 42.9 | 14.3 | 42.6 | 26.7 | 38.0 | 33.5 | 21.8 | 30.6 |
| PROB (Zohar et al., 2023b) | 19.4 | 59.5 | 17.4 | 55.7 | 32.2 | 44.0 | 19.6 | 43.0 | 22.2 | 36.0 | 35.7 | 18.9 | 31.5 |
| CAT (Ma et al., 2023a) | 23.7 | 60.0 | 19.1 | 55.5 | 32.7 | 44.1 | 24.4 | 42.8 | 18.7 | 34.8 | 34.4 | 16.6 | 29.9 |
| RandBox (Wang et al., 2023c) | 10.6 | 61.8 | 6.3 | - | - | 45.3 | 7.8 | - | - | 39.4 | - | - | 35.4 |
| EO-OWOD* (Sun et al., 2024) | 24.6 | 61.3 | 26.3 | 55.5 | 38.5 | 47.0 | 29.1 | 46.7 | **30.6** | 41.3 | 42.4 | 24.3 | 37.9 |
| Hyp-OW (Doan et al., 2024) | 23.5 | 59.4 | 20.6 | - | - | 44.0 | 26.3 | - | - | 36.8 | - | - | 33.6 |
| MEPU-FS (Fang et al., 2023) | 31.6 | 60.2 | 30.9 | 57.3 | 33.3 | 44.8 | 30.1 | 42.6 | 21.0 | 35.4 | 34.8 | 19.1 | 30.9 |
| SGROD (He et al., 2024) | 34.3 | 59.8 | 32.6 | 56.0 | 32.3 | 44.9 | 32.7 | 42.8 | 22.4 | 36.0 | 35.5 | 18.5 | 31.2 |
| SKDF (Ma et al., 2024) | 39.0 | 56.8 | 36.7 | 52.3 | 28.3 | 40.3 | 36.1 | 36.9 | 16.4 | 30.1 | 31.0 | 14.7 | 26.9 |
| KTCN (Xi et al., 2024) | 41.5 | 60.2 | 38.6 | 55.8 | 36.3 | 46.0 | 39.7 | 43.5 | 22.1 | 36.4 | 35.1 | 16.2 | 30.4 |
| ovow* (Li et al., 2024) | 73.5 | 72.1 | 77.5 | 72.4 | **51.0** | 61.7 | 76.1 | 61.6 | **41.6** | 54.9 | **56.0** | 34.3 | **50.6** |
| Zero-Shot (GT) | 59.4 | 69.3 | 56.0 | 69.3 | 41.8 | 55.5 | 55.7 | 55.5 | 31.4 | 47.5 | 47.5 | 26.1 | 42.1 |
| **Ours: OW-VAP** | **58.8** | 68.8 | 56.3 | 68.8 | **42.5** | **55.6** | 55.1 | 55.7 | 29.9 | 47.1 | 47.3 | 25.8 | 42.0 |
| **Ours: OW-VAP*** | **86.6** | **74.5** | **86.8** | **75.6** | 50.1 | **62.3** | **84.8** | **62.4** | 40.1 | **55.0** | 55.2 | **34.8** | 50.1 |
| OW-DETR (Gupta et al., 2022) | 5.7 | 71.5 | 6.2 | 62.8 | 27.5 | 43.8 | 6.9 | 45.2 | 24.9 | 38.5 | 38.2 | 28.1 | 33.1 |
| CAT (Ma et al., 2023a) | 24.0 | 74.2 | 23.0 | 67.6 | 35.5 | 50.7 | 24.6 | 51.2 | 32.6 | 45.0 | 45.4 | 35.1 | 42.8 |
| PROB (Zohar et al., 2023b) | 17.6 | 73.4 | 22.3 | 66.3 | 36.0 | 50.4 | 24.8 | 47.8 | 30.4 | 42.0 | 42.6 | 31.7 | 39.9 |
| EO-OWOD (Sun et al., 2024) | 24.6 | 71.6 | 27.9 | 64.0 | 39.9 | 51.3 | 31.9 | 52.1 | 42.2 | 48.8 | 48.7 | 38.8 | 46.2 |
| Hyp-OW (Doan et al., 2024) | 23.9 | 72.7 | 23.3 | - | - | 50.6 | 25.4 | - | - | 46.2 | - | - | 44.8 |
| MEPU-FS (Fang et al., 2023) | 37.9 | 74.3 | 35.8 | 68.0 | 41.9 | 54.3 | 35.7 | 50.2 | 38.3 | 46.2 | 43.7 | 33.7 | 41.2 |
| SGROD (He et al., 2024) | 48.0 | 73.2 | 48.9 | 64.7 | 36.7 | 50.0 | 47.7 | 47.4 | 32.4 | 42.4 | 42.5 | 32.6 | 40.0 |
| SKDF (Ma et al., 2024) | 60.9 | 69.4 | 60.0 | 63.8 | 26.9 | 44.4 | 58.6 | 46.2 | 28.0 | 40.1 | 41.8 | 29.6 | 38.7 |
| ovow (Li et al., 2024) | 71.3 | 76.4 | 74.4 | 75.0 | 59.8 | 67.0 | 74.6 | 67.0 | 53.8 | 62.6 | 65.5 | 56.9 | 63.4 |
| Zero-Shot (GT) | 87.3 | 79.0 | 86.5 | 79.0 | 58.4 | 68.2 | 86.3 | 68.2 | 56.7 | 64.4 | 64.4 | 58.0 | 62.8 |
| **Ours: OW-VAP** | **82.9** | **79.6** | **85.6** | **79.8** | **62.3** | **70.6** | **85.7** | **70.7** | **57.6** | **66.3** | **66.4** | **59.1** | **64.6** |

soft labels to weight the pseudo-label loss:

$$\mathcal{L}_{pseudo} = \text{mean}_{|\text{T}(E_{vis}^*)|}\text{W}(\text{T}(E_{vis}^*))\text{CE}(\text{T}(E_{vis}^*)|1),$$

$$\text{T}(E_{vis}^*) = \sigma(\gamma \cdot \max_{|Att|}\text{Sim}(\text{M}(E_{vis}^*), E_{att}) + \beta), \quad (9)$$

where $\text{CE}(\cdot\,|1)$ denotes the cross-entropy loss for the current prediction when the target is 1. The parameters $\gamma$ and $\beta$ are learnable scaling factors, both being scalars. $E_{vis}^*$ and $\text{T}(\cdot)$ represent the selected pseudo-label visual embeddings and VAP, respectively. $\sigma$ and M denote the Sigmoid function and Multilayer Perceptron (MLP). For other samples, we simply the apply cross-entropy to calculate loss (Appendix E).

### 3.4. Unknown Inference

Once the VAP is trained, we extract attributes from all visual regions to estimate the probability of unknown classes. To avoid confusion between known and unknown objects, consistent with prior work (Zohar et al., 2023a), we simply add the out-of-distribution probability to balance the model. Thus, for a given visual embedding $e_{vis}$, the corresponding

probability of the unknown class is denoted by:

$$P_u(e_{vis}) = (1 - \max_{|c|}\text{Sim}(e_{vis}, E_c))\cdot$$
$$(b \cdot \text{mean}_{|Att|}\text{Sim}(e_{vis}, E_{att}) + (1 - b)\text{T}(e_{vis})). \quad (10)$$

## 4. Experiments

Details about the dataset, evaluation metrics, and implementation can be found in the Appendix C, Appendix D and Appendix E.

### 4.1. Main Results

We compare OW-VAP with previous state-of-the-art (SOTA) methods on the OWOD standard benchmarks, MOWODB and SOWODB. The results are shown in Table 1. Further comparisons can be found in Appendix G. Due to data leakage in ORE(Joseph et al., 2021), we follow previous work (Wang et al., 2023c; Doan et al., 2024) by using the energy model-excluded version (ORE-EBUI) for evaluation. EO-OWOD performs deduplication on the MOWODB test

*Table 2.* **Detection Accuracy of Unknown Classes and Confusion with Known Classes on the MOWODB Benchmark.** U-AP denotes average precision for unknown classes, which is a widely used metric in object detection. WI and A-OSE indicate the confusion between known and unknown classes in the open set. WI represents the loss in known class precision after introducing unknown class detection. A-OSE indicates the absolute number of unknown objects predicted as known classes. * denotes results after deduplication of test set.

| Task IDs (→) | Task 1 | | | | Task 2 | | | | Task 3 | | | |
|---|---|---|---|---|---|---|---|---|---|---|---|---|
| Method | U-Recall (↑) | U-AP (↑) | WI (↓) | A-OSE (↓) | U-Recall (↑) | U-AP (↑) | WI (↓) | A-OSE (↓) | U-Recall (↑) | U-AP (↑) | WI (↓) | A-OSE (↓) |
| ORE (Joseph et al., 2021) | 5.7 | 0.7 | 0.0621 | 10459 | 2.7 | 0.1 | 0.0282 | 10445 | 2.3 | 0.1 | 0.0211 | 7990 |
| SA (Yang et al., 2021) | 1.9 | 0.2 | 0.0563 | 23320 | 0.8 | 0.0 | 0.0181 | 16768 | 0.1 | 0.0 | 0.0136 | 1428 |
| ALLOW (Ma et al., 2023b) | 13.6 | 4.9 | 0.0564 | 45689 | 10.0 | 0.7 | 0.0274 | 24709 | 14.3 | 0.4 | 0.0194 | 14952 |
| OW-DETR (Gupta et al., 2022) | 7.7 | 0.1 | 0.0599 | 42331 | 5.8 | 0.0 | 0.0319 | 25857 | 6.0 | 0.0 | 0.0220 | 18056 |
| PROB (Zohar et al., 2023b) | 19.2 | 1.6 | 0.0574 | 5238 | 17.1 | 0.4 | 0.0342 | 6556 | 19.0 | 0.4 | 0.0154 | 2732 |
| EO-OWOD* (Sun et al., 2024) | 24.7 | 2.6 | 0.0303 | 4163 | 26.7 | 0.2 | 0.0097 | 1800 | 32.1 | 0.1 | 0.0075 | 1509 |
| SGROD (He et al., 2024) | 33.2 | 3.6 | 0.0450 | 4567 | 31.8 | 2.5 | 0.0279 | 2624 | 31.7 | 0.3 | 0.0152 | 1549 |
| SKDF (Ma et al., 2024) | 39.0 | 1.5 | 0.0698 | 13693 | 36.8 | 1.3 | 0.0279 | 7829 | 36.1 | 0.8 | 0.0168 | 5142 |
| KTCN (Xi et al., 2024) | 41.5 | 1.2 | 0.0809 | 13368 | 38.6 | 0.9 | 0.0439 | 11861 | 39.7 | 0.5 | 0.0250 | 6382 |
| ovow* (Li et al., 2024) | 73.5 | 10.4 | 0.0175 | 1038 | 77.5 | 10.2 | 0.0047 | 529 | 76.1 | 9.4 | 0.0030 | 448 |
| Zero-Shot (GT) | 59.4 | 24.9 | 0.0213 | 15761 | 56.0 | 21.1 | 0.0150 | 13978 | 55.7 | 19.1 | 0.0114 | 12185 |
| **Ours: OW-VAP** | **58.8** | **13.0** | **0.0185** | **996** | **56.3** | **9.3** | **0.0082** | **531** | **55.1** | **7.8** | **0.0059** | **443** |
| **Ours: OW-VAP*** | **86.6** | **18.9** | 0.0194 | **983** | **86.8** | **14.1** | 0.0085 | **523** | **84.8** | **11.8** | 0.0059 | **435** |

*Table 3.* **Ablation Study on MOWODB Benchmark.** Base Model refers to the OVD detector utilizing only known class names. General Prompt denotes the use of generalized attribute descriptions, and OOD Prob indicates the out-of-distribution probability. VAP significantly enhances the detector's recall capability for unknowns, while PSLA focuses on improving the precision of detecting unknowns.

| Task IDs (→) | Task 1 | | | | Task 2 | | | | Task 3 | | | |
|---|---|---|---|---|---|---|---|---|---|---|---|---|
| Method | U-Recall (↑) | U-AP (↑) | WI (↓) | A-OSE (↓) | U-Recall (↑) | U-AP (↑) | WI (↓) | A-OSE (↓) | U-Recall (↑) | U-AP (↑) | WI (↓) | A-OSE (↓) |
| Base Model | 0.0 | 0.0 | 0.0332 | 30068 | 0.0 | 0.0 | 0.0247 | 25685 | 0.0 | 0.0 | 0.0189 | 21816 |
| + General Prompt + OOD Prob | 45.6 | 10.4 | 0.0216 | 17135 | 44.2 | 7.5 | 0.0157 | 14298 | 45.7 | 6.1 | 0.0137 | 13925 |
| + Visual Attribute Parser | 58.5 | 10.5 | 0.0210 | 1025 | 55.7 | 6.7 | 0.0087 | 544 | 54.8 | 6.2 | 0.0061 | **438** |
| + Probabilistic Soft Label | **58.8** | **13.0** | **0.0185** | **996** | **56.3** | **9.3** | **0.0082** | **531** | **55.1** | **7.8** | **0.0059** | 443 |

set, resulting in a reduction of 1k test images and 5k test instances. To ensure a fair comparison, we introduce a deduplicated test version, OW-VAP*, to represent the performance. Clearly, OW-VAP surpasses previous SOTA by a significant margin. Compared to methods without the foundation model, we achieve double the performance of CAT (Ma et al., 2023a) and PROB (Zohar et al., 2023b). Against methods utilizing the foundation model, our approach also demonstrates substantial performance advantages. When compared to SAM-based methods like SGROD (He et al., 2024) and KTCN (Xi et al., 2024), we lead by 17.9 U-Recall. Compared to OVD detector methods, such as SKDF (Ma et al., 2024), we achieve a 20.4 U-Recall advantage. Notably, in Task 2, we surpass the generalization upper bound of the OVD detector (Zero-Shot (GT)) by +0.3 U-Recall. Against deduplicated SOTA methods, ovow (Li et al., 2024), we also achieve significant improvements, with a 13.1 U-Recall advantage in Task 1. In SOWODB, we exceed ovow with a recall advantage of over 10 U-Recall, while maintaining unknown recall close to zero-shot testing.

## 4.2. Comparison of Accuracy and Confusion

To better demonstrate the detector's ability to identify unknown classes, we use a more stringent metric, U-AP, which is the most widely used metric for object detection.. Consistent with prior work (Zohar et al., 2023b; Ma et al., 2023b), we also employ WI and A-OSE to assess the confusion between unknown and known classes. The results are shown in Table 2. To obtain U-AP, we first extract relevant values from ALLOW (Ma et al., 2023b). For the remaining U-AP values, we use the official code and download the corresponding weights for evaluation. Note that in official updates, the detector's performance may differ from what is presented in the paper. OW-VAP exhibits significant performance advantages. In U-AP, OW-VAP leads across all tasks, achieving a maximum gap of 8.5 U-AP in Task 1. Additionally, in Task 1, we achieved 13.0 U-AP, falling short of the OVD generalization upper limit by only 11.9 U-AP. In the WI comparison, OW-VAP shows marked improvement over previous methods, lagging slightly behind ovow only in the

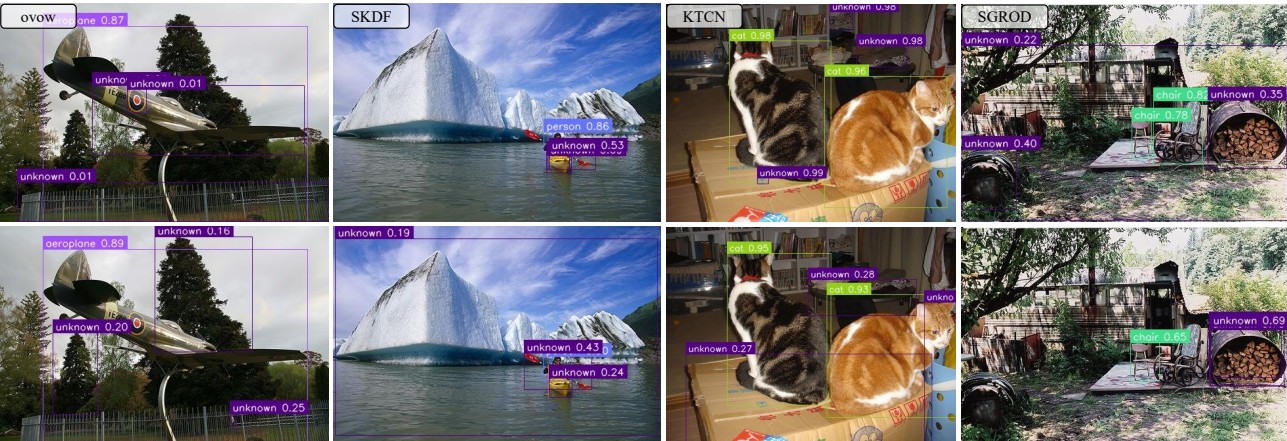

*Figure 5.* **Qualitative Comparison with Recent SOTA using Foundation Model on MOWODB.** For a fair evaluation, we rank predictions by unknown confidence and display only top three. The results of our OW-VAP are shown in the second row.

deduplicated comparison. In the A-OSE comparison, OW-VAP surpasses all other methods, even outperforming the deduplicated ovow* in Task 1. Furthermore, in the unknown recall comparison, OW-VAP achieves a performance gap of over 10. Overall, OW-VAP not only surpasses other SOTA methods in its ability to detect unknown classes but also maintains, and in some cases, exceeds, their performance in distinguishing between known and unknown classes.

### 4.3. Ablation Study

To validate the effectiveness of the proposed components, we conducted incremental ablation experiments as shown in Table 3. For details on hyperparameter ablation, please refer to the Appendix F. Upon introducing attribute descriptions and out-of-distribution probability, the detector's recall and average precision improve to 45.6 and 10.4, respectively, demonstrating the OVD detector's strong generalization capability. However, confusion between known and unknown objects remains high. With the integration of VAP, which learns to parse visual region attributes, recall is significantly improved. In Task 1, VAP achieves an increase of 12.9 in U-Recall. Nevertheless, in subsequent incremental experiments, VAP slightly lags in U-AP. In Task 2, VAP falls behind the coarse attribute method by 0.8 in U-AP. Finally, PSLA distinguishes positive and negative samples during training, which significantly enhances the model's detection performance. It achieves the highest detection precision and recall, with 58.8 U-Recall and 13.0 U-AP in Task 1. In addition, PSLA demonstrates robust performance in incremental learning, where it not only maintains a consistent advantage in U-Recall but also shows substantial improvements in U-AP. Moreover, it maintains an advantage in incremental learning, achieving notable improvements in recall while enhancing 2.6 and 1.6 U-AP in Tasks 2 and 3, respectively.

### 4.4. Visualization

We performed a qualitative comparison of SOTA methods using the foundation model, with results presented in Figure 5. OW-VAP demonstrates strong capability in detecting unknown classes. Compared to SKDF, our method correctly identifies objects in the background, such as boats, paddles, and icebergs, whereas SKDF redundantly predicts the boat and overlooks other objects. Comparison with ovow, OW-VAP successfully detects trees in the background and small-sized individuals beneath the aircraft. ovow, however, erroneously identifies the circle within the aircraft as a potential object, indicating lower detection precision. Similar to ovow, KTCN misclassifies paintings within objects as potential targets, an error not present in our OW-VAP. SGROD, like SKDF, repeatedly predicts the same potential object, whereas OW-VAP accurately recognizes other background objects, detecting an abandoned car and tire. Overall, OW-VAP exhibits superior recall capability and detection precision for unknown objects.

## 5. Conclusion

In this paper, we propose a novel network architecture, OW-VAP. To identify unknown objects, we propose a Visual Attribute Parser (VAP) that analyzes the attributes of visual regions and assesses the likelihood of these regions containing unknown objects. We select samples from background regions that exhibit attribute similarity above the average level and label them as pseudo labels. Furthermore, we propose the Probabilistic Soft Label Assignment (PSLA) to mitigate the optimization conflicts arising from background noise in pseudo labels. In evaluations of OWOD, OW-VAP demonstrates performance that approaches or exceeds the generalization capability limits. We anticipate that OW-VAP will advance the application of OWOD in real world.

# Impact Statement

This paper presents work whose goal is to advance the field of Machine Learning. There are many potential societal consequences of our work, none which we feel must be specifically highlighted here.

# Acknowledgements

The authors from Ant Group are supported by the Leading Innovative and Entrepreneur Team Introduction Program of Hangzhou (Grant No.TD2022005) and National Key Research and Development Program of China (Grant No. 2024YFE0105400). In addition, this work was supported by Ant Group Research Intern Program. We would like to thank all the anonymous reviewers for their insightful comments.

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

*Table 4.* **Detailed Attribute Descriptions.** Type indicates the attribute type, identifying the object features of interest. Description provides the text description corresponding to the type code. For simplicity, we use only nine attributes.

| Type | Description |
| --- | --- |
| Shape | The object has a distinctive shape |
| Color | The object has a noticeable color |
| Texture | The surface of the object has a particular texture |
| Size | The object has a specific size |
| Context | The object is found in certain environments |
| Features | The object exhibits characteristic features |
| Appearance | The object has a unique appearance |
| Behavior | The object is used in various ways |
| Environment | The object is associated with a certain type of environment |
| Material | The object is made from a specific material |

## A. Detailed and Coarse Attributes

Table 4 presents the coarse textual attributes we used. FOMO (Zohar et al., 2023a) utilizes a large language model (LLM), GPT 3.5, to generate attributes for known categories. These attributes encompass nine types, corresponding to the nine types in Table 4. For each type, the LLM lists the relevant attribute options. For example, for the attribute type color, the LLM lists options such as blue, which are then combined into descriptions like: object which color (is/has/etc) blue. FOMO expects potential objects to exhibit similar attributes, meaning its performance is constrained by the LLM descriptions. In contrast, we use coarse attributes. Unlike FOMO, which seeks fine-grained attributes, we focus on whether a particular attribute of potential objects is prominent. Therefore, we use terms describing degree to represent attribute types. For instance, for the attribute type Color, we provide the description: The object has a noticeable color. By using these representations, we no longer rely on the LLM to generate attributes, thereby avoiding performance limitations on the detector.

## B. Problem Definition

Open World Object Detection (OWOD) consists of two main components: detecting unknown objects and incremental learning. To accurately simulate this process, the detection task is divided into a series of subtasks $T = \{T_1, T_2, ..., T_{|T|}\}$. Given a dataset $D = \{X, Y\}$, where $Y$ represents the labels corresponding to the image set $X$, the label categories are partitioned into subsets $\mathcal{K} = \{\mathcal{K}_1, \mathcal{K}_2, ..., \mathcal{K}_{|\mathcal{K}|}\}$. In $T_1$, a subset $D_1 = \{X_1, Y_1\}$ is extracted from the dataset $D$. Annotations belonging to the category subset $\mathcal{K}_1$ are selected from $Y_1$ to simulate a partially annotated scenario, $\hat{Y}_1 = \{y|\ y \in Y_1 \wedge\ y \in \mathcal{K}_1\}$. During testing, the detector is required to detect all classes in $\mathcal{K}$, with unannotated classes in subset $\mathcal{K}$ considered as the unknown classes of interest. In subtask $T_2$, another subset $D_2 = \{X_2, Y_2\}$ is obtained. Unlike before, the current set of classes $\hat{Y}_2$ excludes both the unknown classes and the annotations belonging to $\hat{Y}_1$. $T_2$ is referred to as incremental learning, which requires detector to fine-tune its existing knowledge (trained in $T_1$) to adapt to new classes.

## C. Dataset Details

Table 5 presents the detailed partitioning of the OWOD benchmark. MOWODB (Table 5(a)) combines the VOC (Everingham et al., 2010) and COCO (Lin et al., 2014) datasets. In MOWODB, all VOC categories are regarded as Task 1, while the remaining categories in COCO are regarded as unknown classes. During testing, the validation sets of VOC and COCO are mixed to form the test set of MOWODB. When evaluating, all the seen categories will be treated as known, and then these known categories will be divided into the categories introduced by the previous tasks and the categories introduced by the current task. In the subsequent tasks, the division is carried out according to the superclasses of COCO. For example, both hair drier and toothbrush belong to the same superclass indoor, and they will be introduced as newly annotated instances in Task 4. SOWODB (Table 5(b)) only uses the COCO dataset and has a smaller amount of data compared to MOWODB. Without the inclusion of VOC categories, SOWODB exhibits more isolated semantic divisions. In MOWODB, detectors are exposed to other supercategories earlier due to VOC categories being subsets of other supercategories. It is noteworthy that the original ORE implementation included duplicate images in the test set, resulting in higher test counts than defined in MOWODB. For instance, in Task 1, there are 23,320 unknown instances. In the EO-OWOD test, duplicate instances were removed, and ovow followed this procedure. Thus, in OWOD evaluations, we include a comparison of results after deduplication.

## D. Metric

OWOD, as a special task of OD, requires the detector to adapt to the open-world environment. For evaluating performance on unknown objects, OD's conventional evaluation metrics can be employed. However, due to the lack of information on unknown objects, directly applying metrics like Average Precision (AP) results in similar performance across all methods. For instance, when evaluated with AP, methods such as ORE (Joseph et al., 2021), OW-DETR (Gupta et al., 2022), and PROB (Zohar et al., 2023b) exhibit comparable performance. Therefore, before employing the foundation models, researchers use compromise metrics to assess specific capabilities of the model, rather than directly using AP to evaluate overall performance. U-Recall represents the recall rate of unknown classes; an unknown class

*Table 5.* **Detailed Description of Dataset Partitioning.** Train and test images denote the number of images in the training and test sets, respectively, while the corresponding instances represent the number of test instances. MOWODB utilizes a combination of VOC and COCO datasets, whereas SOWODB exclusively uses the COCO dataset.

*(a).* Semantic category division of MOWODB.

| MOWODB | Task 1 | Task 2 | Task 3 | Task 4 |
|---|---|---|---|---|
| Split | VOC Classes | Outdoor, Accessories Appliances, Truck | Sports, Food | Electronic, Indoor, Kitchen, Furniture |
| train images | 16551 | 45520 | 39402 | 40260 |
| test images | 4952 | 1914 | 1642 | 1738 |
| train instances | 47223 | 113741 | 114452 | 138996 |
| test instances | 14976 | 4966 | 4826 | 6039 |

*(b).* Semantic category division of SOWODB.

| SOWODB | Task 1 | Task 2 | Task 3 | Task 4 |
|---|---|---|---|---|
| Split | Animals, Persion Vehicles | Appliances, Outdoor Accessories, Furniture | Sports, Kitchen, Food | Indoor, Electronic |
| train images | 89490 | 55870 | 39402 | 38903 |
| test images | 3793 | 2351 | 1642 | 1691 |
| train instances | 421243 | 163512 | 114452 | 160794 |
| test instances | 17786 | 7159 | 4826 | 7010 |

*Table 6.* **Hyperparameter Ablation Analysis.** The three sub-tables below show the impact of hyperparameters on model performance. The selected hyperparameters are indicated in bold. All analyses are conducted on MOWODB. To reduce the number of parameters, we set the threshold $\alpha$ (Equation (4)) to the inference threshold of 0.01 used in YOLO-World.

*(a).* The impact of $\delta$ (Equation (5)).

| Setting $\delta$ | U-Recall ($\uparrow$) | U-AP ($\uparrow$) | WI ($\uparrow$) | A-OSE ($\uparrow$) |
|---|---|---|---|---|
| 0 | 58.6 | 7.5 | 0.0221 | 1376 |
| 5 | 58.7 | 9.5 | 0.0193 | 1212 |
| **10** | **58.6** | **10.6** | **0.0182** | **1167** |
| 15 | 58.8 | 10.2 | 0.0184 | 1135 |
| 20 | 58.8 | 9.9 | 0.0173 | 1111 |
| 25 | 58.2 | 9.8 | 0.0176 | 1132 |
| 30 | 58.3 | 9.5 | 0.0175 | 1093 |
| 35 | 58.4 | 9.6 | 0.0177 | 1100 |
| 40 | 58.7 | 9.6 | 0.0173 | 1087 |
| 45 | 58.5 | 9.5 | 0.0180 | 1081 |

*(b).* The impact of $n$ (Equation (6)).

| Setting $n$ | U-Recall ($\uparrow$) | U-AP ($\uparrow$) | WI ($\uparrow$) | A-OSE ($\uparrow$) |
|---|---|---|---|---|
| 1 | 58.4 | 10.7 | 0.0180 | 995 |
| 2 | 58.1 | 10.4 | 0.0183 | 1059 |
| **3** | **58.4** | **11.0** | **0.0182** | **969** |
| 4 | 58.4 | 10.4 | 0.0177 | 1015 |
| 5 | 58.4 | 10.7 | 0.0173 | 1025 |
| 6 | 58.7 | 10.8 | 0.0178 | 985 |
| 7 | 58.7 | 10.6 | 0.0175 | 1022 |
| 8 | 58.1 | 10.4 | 0.0176 | 1009 |
| 9 | 58.5 | 10.5 | 0.0177 | 988 |
| 10 | 58.6 | 9.2 | 0.0221 | 1266 |

*(c).* The impact of $b$ (Equation (10)).

| Setting $b$ | U-Recall ($\uparrow$) | U-AP ($\uparrow$) | WI ($\uparrow$) | A-OSE ($\uparrow$) |
|---|---|---|---|---|
| 0.1 | 58.3 | 11.6 | 0.0235 | 3094 |
| 0.2 | 58.8 | 12.0 | 0.0214 | 2007 |
| 0.3 | 58.9 | 12.4 | 0.0200 | 1533 |
| 0.4 | 59.0 | 12.5 | 0.0193 | 1272 |
| 0.5 | 58.9 | 12.7 | 0.0187 | 1108 |
| **0.6** | **58.8** | **13.0** | **0.0185** | **996** |
| 0.7 | 58.8 | 13.1 | 0.0185 | 915 |
| 0.8 | 58.6 | 13.2 | 0.0180 | 831 |
| 0.9 | 58.6 | 13.4 | 0.0177 | 770 |
| 1.0 | 58.5 | 13.2 | 0.0178 | 728 |

is considered recalled if the IOU between the prediction and the unknown class ground truth exceeds 0.5. WI (Dhamija et al., 2020) represents the relative open-set error. It is composed of the ratio of the loss of detection accuracy of the known classes after the introduction of the unknown classes:

$$WI = \frac{P_K}{P_{K \cup U}} - 1, \qquad (11)$$

where $P_K$ and $P_{K \cup U}$ represent the accuracy of known classes and the overall accuracy in open-world scenarios, respectively. Consistent with previous methods (Joseph et al., 2021; Ma et al., 2023b), we calculate this value using an IoU threshold of 0.8. Unlike WI, A-OSE calculates the absolute value directly. A-OSE directly measures the number of overlaps between known predictions and unknown class labels. Specifically, A-OSE is incremented by one when the IoU between a known prediction and an unknown class exceeds 0.5.

These metrics act as a trade-off solution in scenarios where the performance of early OWOD detectors does not meet expectations. Initially, these detectors faced challenges in ac-

curately identifying and recalling unknown objects, leading to a need for a pragmatic approach in performance assessment. However, with the introduction of foundation models, which incorporate more advanced algorithms and learning capabilities, the detectors have now approached the maximum achievable performance in terms of unknown object recall. This advancement necessitates a more rigorous evaluation process to differentiate between subtle improvements in detection capabilities. Consequently, we include the Unknown Average Precision (U-AP) metric during evaluation as a more stringent criterion.

## E. Implementation Details

**Training of positive and negative samples in VAP.** For positive and negative samples, the loss is calculated similarly to the pseudo-labels, using cross-entropy:

$$\mathcal{L}_{positive} = \text{mean}_{|\text{T}(E_{vis}^+)|} \text{CE}(\text{T}(E_{vis}^+)|1), \qquad (12)$$

where $E_{vis}^+$ represents the visual embedding corresponding to the positive samples generated by the matching algorithm,

*Table 7.* **Performance Comparison on the OWOD Standard Evaluation Benchmark.** The top section shows MOWODB, while the bottom section shows SOWODB. FOMO-ZS+ denotes the use of a generic description, object, as a cue for unknown objects, combined with OOD scores to estimate the likelihood of unknown classes. FOMO-ZS+IN uses ImageNet categories, removing unknown class names and using the remaining class names as cues for unknown objects. FOMO refers to the use of all attributes in FOMO.

| Task IDs (→) | Task 1 | | Task 2 | | | | Task 3 | | | | Task 4 | | |
|---|---|---|---|---|---|---|---|---|---|---|---|---|---|
| | U-Recall | mAP (↑) | U-Recall | mAP(↑) | | | U-Recall | mAP(↑) | | | mAP(↑) | | |
| Method | (↑) | Current Known | (↑) | Previously Known | Current Known | Both | (↑) | Previously Known | Current Known | Both | Previously Known | Current Known | Both |
| FOMO-ZS+ | 24.0 | 68.1 | 25.4 | 68.5 | 41.7 | 55.1 | 28.6 | 55.2 | **31.3** | 47.2 | **47.5** | **26.1** | **42.1** |
| FOMO-ZS+IN | 52.4 | 67.8 | 50.1 | 67.7 | 41.3 | 54.5 | 49.9 | 54.6 | 31.2 | 46.8 | 46.8 | 25.0 | 41.4 |
| FOMO | 54.6 | 67.4 | 51.9 | 59.0 | 37.4 | 48.2 | 52.1 | 42.8 | 24.4 | 36.7 | 35.0 | 21.4 | 35.0 |
| **Ours: OW-VAP** | **58.8** | **68.8** | **56.3** | **68.8** | **42.5** | **55.6** | **55.1** | **55.7** | 29.9 | 47.1 | 47.3 | 25.8 | 42.0 |
| FOMO-ZS+ | 31.4 | 78.5 | 38.6 | 78.8 | 58.2 | 68.0 | 44.0 | 68.0 | 56.5 | 64.2 | 64.3 | 58.0 | 62.8 |
| FOMO-ZS+IN | 77.4 | 77.6 | 78.4 | 77.6 | 58.8 | 67.7 | 78.6 | 67.7 | 56.2 | 63.9 | 63.8 | 57.3 | 62.2 |
| FOMO | 78.4 | 69.9 | 80.5 | 60.0 | 50.1 | 54.8 | 80.8 | 50.7 | 45.4 | 49.0 | 50.3 | 45.9 | 49.2 |
| **Ours: OW-VAP** | **82.9** | **79.6** | **85.6** | **79.8** | **62.3** | **70.6** | **85.7** | **70.7** | **57.6** | **66.3** | **66.4** | **59.1** | **64.6** |

*Table 8.* **Detection Accuracy of Unknown Classes and Confusion with Known Classes on the MOWODB Benchmark.** All implementations in the FOMO series are consistent with OW-VAP by assigning samples with unknown probabilities greater than known probabilities to the unknown class. Our OW-VAP demonstrates a significant lead in both detection precision and recall.

| Task IDs (→) | Task 1 | | | | Task 2 | | | | Task 3 | | | |
|---|---|---|---|---|---|---|---|---|---|---|---|---|
| Method | U-Recall (↑) | U-AP (↑) | WI (↓) | A-OSE (↓) | U-Recall (↑) | U-AP (↑) | WI (↓) | A-OSE (↓) | U-Recall (↑) | U-AP (↑) | WI (↓) | A-OSE (↓) |
| FOMO-ZS+ | 24.0 | 6.8 | 0.0295 | 26353 | 25.4 | 5.9 | 0.0219 | 22172 | 28.6 | 5.5 | 0.0174 | 19528 |
| FOMO-ZS+IN | 52.4 | 3.5 | 0.0221 | 9521 | 50.1 | 2.4 | 0.0152 | 8077 | 49.9 | 3.1 | 0.0107 | 6138 |
| FOMO | 54.6 | 4.9 | 0.0564 | 1937 | 51.9 | 2.6 | 0.0103 | **332** | 52.1 | 2.1 | 0.0065 | **196** |
| **Ours: OW-VAP** | **58.8** | **13.0** | **0.0185** | **996** | **56.3** | **9.3** | **0.0082** | 531 | **55.1** | **7.8** | **0.0059** | 443 |

i.e., $(S > 0)$. Similarly, the loss for negative samples is:

$$\mathcal{L}_{negative} = \text{mean}_{|\text{T}(E^-_{vis})|}\text{CE}(\text{T}(E^-_{vis})|0), \quad (13)$$

where $E^-_{vis}$ represents the visual embedding corresponding to the negative samples, i.e. $(S = 0)$. Therefore, the final loss for training VAP is: $\mathcal{L} = \mathcal{L}_{positive} + \mathcal{L}_{pseudo} + \mathcal{L}_{negative}$.

**Setting.** Our approach is based on the OVD detector, YOLO-World (Cheng et al., 2024). To ensure a fair comparison with ovow, we use the XL version. The object attributes are described using nine categories, generating only nine distinct sentences. For the text encoder, we use the CLIP model, specifically the version clip-vit-base-patch32. During training, we freeze the text and visual encoders, training only the VAP and known class embeddings. All experiments are conducted using 8 V100 GPUs (total 128 GB). We implement all experiments using MMDetection (Chen et al., 2019). For the experimental parameters, we follow the official settings of YOLO-World, altering only the training epochs: 10 epochs for MOWODB and 2 for SOWODB.

**Inference.** During inference, we observed that the detector often assigns different class IDs to the same sample, which increases the difficulty of distinguishing between positive and negative samples. Therefore, for the same sample, if the known confidence is lower than the unknown confidence, we set the known confidence to 0. In the SOWODB, such a situation did not occur. As a result, we only modify the confidence in MOWODB, while disregarding SOWODB.

**Incremental learning.** Since we mine latent objects in Task 1, VAP exhibits strong generalization capability. Fine-tuning it during subsequent incremental learning would harm the detector's existing knowledge, as previously seen classes are labeled as background. Therefore, we only fine-tune VAP during the knowledge replay phase. The replay process is conducted exclusively on the MOWODB, while in the SOWODB, we directly use the VAP from Task 1 to predict all unknown classes, bypassing the fine-tuning process.

**Code.** In the near future, once the code passes the company's review, we will release all the trained code, weights, and models (including other SOTA) along with visualization

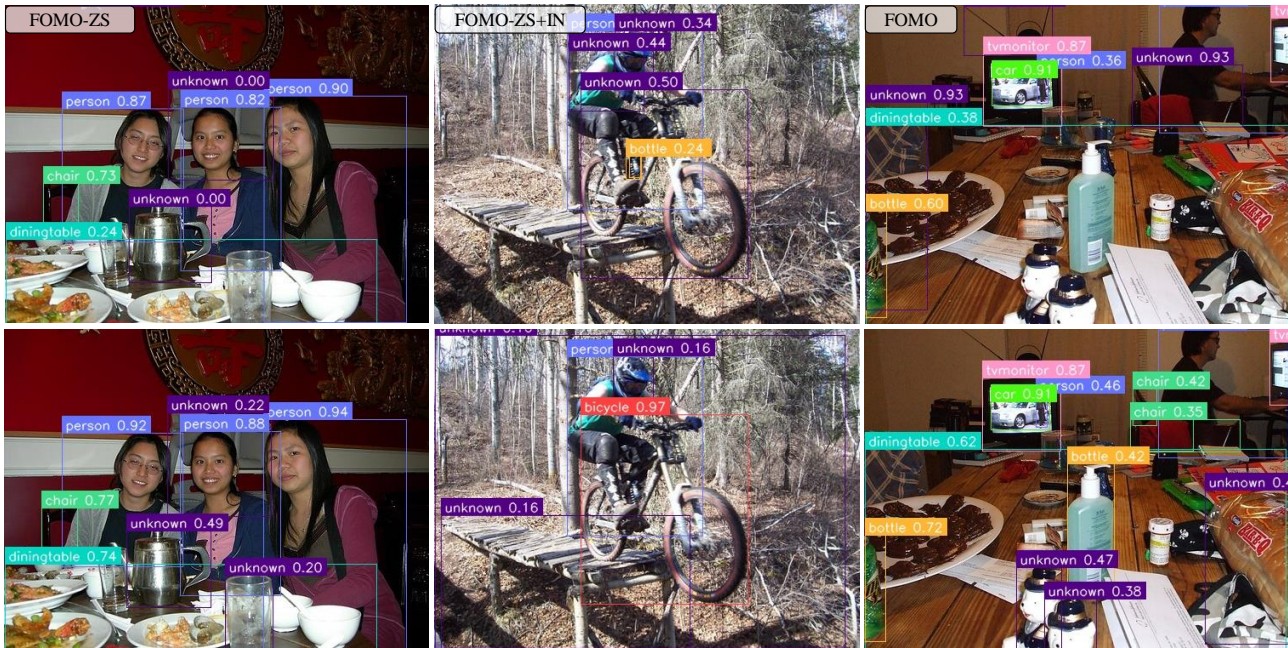

*Figure 6.* **Qualitative Comparison with FOMO Series on MOWODB.** As before, for a fair evaluation, we rank predictions by unknown confidence and display only the top three. OW-VAP demonstrates superiority in both detection precision and recall.

files at the MOWODB dataset level.

## F. Ablation Experiment of Hyperparameters

$\delta$. In Equation (5), we introduce the hyperparameter $\delta$. This parameter controls the number of samples selected from the background region. Intuitively, the more samples selected, the higher the recall rate, since more potential samples are marked with pseudo-labels. However, this also increases the risk of background samples being incorrectly labeled as pseudo-labels. As shown in Table 6(a), increasing the number of recalls enhances the unknown recall rate, reaching a maximum of 58.8 U-Recall when is set to 20. However, further increasing $\delta$ does not improve recall, indicating that background samples in the pseudo-labels negatively affect the detector's ability to identify unknown objects. Therefore, we set $\delta$ to 10 for subsequent experiments based on the maximum U-AP value.

$n$. In Equation (6), we introduce the parameter $n$. This parameter is utilized to control the number of parameters in the queue. When establishing a Weibull distribution, estimating the parameters within small batches leads to training instability. To mitigate this, we estimate the parameters from historical batches to stabilize the training. Utilizing a longer queue, however, results in less focus on the current batch. There is a trade-off involved in the length of the queue. As shown in Table 6, increasing the length of the queue decreases both the recall and detection precision of

the detector. Compared to longer queues, using a shorter queue yields better performance (10.7 U-AP with $n = 1$). Therefore, considering the performance trends of the parameters as $n$ varies, we set $n$ to 3 for subsequent experiments.

$b$. In Equation (10), we integrate the raw attributes as a rough estimate of the current visual region. Subsequently, a visual attribute extractor is used for more refined predictions. To balance the contributions of these two components, we introduce the balancing parameter $b$. The performance of the detector as a function of the parameter $b$ is shown in Table 6(c). Overall, when $b$ is set to a small value, the detector exhibits poor detection precision, achieving only 11.6 U-AP at $b = 0.1$. The Visual Attribute Predictor (VAP) further enhances the detection precision. When $b$ is set to 1, effectively excluding the raw attributes, the detector achieves 13.2 U-AP and performs well on both WI and A-OSE. However, this setting leads to a significant decrease in known performance. Therefore, we strike a balance between the contributions of the two components by setting the value of $b$ to 0.6.

## G. More Comparisons

### G.1. Implementation Details of FOMO

As previously mentioned, FOMO relies on attributes generated by a large language model (LLM) to predict unknown objects. We use GPT-3.5 (in its official configuration) to generate attributes for the OWOD benchmark. The similar-

**Algorithm 1** Visual Attribute Parser Pseudocode

**Input:** $Att, Name, I, Y_I, E_{vis}^+, \alpha, \delta$ {Input data: attributes, class names, image, ground truth, positive visual embeddings, thresholds.}

**Output:** $S_{vap}$ {Output: pseudo-label indicator.}

$E_{att} \leftarrow \text{Enc}_t(Att)$ {Encode attributes to embedding.}

$E_c \leftarrow \text{Enc}_t(Name)$ {Encode class names to embedding.}

$E_{vis} \leftarrow \text{Enc}_v(I)$ {Encode image to visual embedding.}

$S \leftarrow \arg\min_\pi \triangle(E_c, E_{vis}, Y_I)$ {Compute matching score.}

$cond1 \leftarrow \max(\text{Sim}(E_{vis}, E_{att}))$
$\qquad\qquad \geq \text{mean}(\max(\text{Sim}(E_{vis}^+, E_{att})))$
{Condition 1: Compare similarity with positive embeddings.}

$cond2 \leftarrow \max(\text{Sim}(E_{vis}, E_{att})) \geq \alpha$
{Condition 2: Compare similarity with threshold $\alpha$.}

$cond3 \leftarrow \text{Top}_\delta(\text{Sim}(E_{vis}, E_{att}))$
{Condition 3: Check top $\delta$ similarities.}

$S_{vap} \leftarrow \mathbb{1}(cond1 \wedge cond2 \wedge cond3)$
{Compute final pseudo-label indicator.}

---

**Algorithm 2** Probabilistic Soft Label Assignment (PSLA)

**Input:** $E_{vis}^+, E_{att}, \alpha, \delta, \gamma, \beta$

**Output:** Soft labels $W(x)$, Pseudo-label loss $\mathcal{L}_{pseudo}$

$\lambda^+ \leftarrow \text{CalculateWeightedAverage}(E_{vis}^+)$
{Calculate the weighted average of positive sample distribution parameters}

$\lambda^- \leftarrow \text{EstimateNegativeDistributionParameters}(E_{vis}^+)$
{Estimate the negative sample distribution}

$Wb^+(x) \leftarrow \text{Wb}(x, \lambda^+, k, \theta)$ {Calculate positive sample distribution using Weibull distribution}

$Wb^-(x) \leftarrow \text{Wb}(x, \lambda^-, k, \theta)$ {Calculate negative sample distribution using Weibull distribution}

$W(x) \leftarrow \frac{Wb^+(x)}{Wb^+(x)+Wb^-(x)}$ {Compute soft labels based on positive and negative distributions}

$S_{vap} \leftarrow \text{GeneratePseudoLabels}(W(x), E_{vis}^+, E_{att}, \alpha, \delta)$
{Generate pseudo-labels using PSLA method}

$\mathcal{L}_{pseudo} \leftarrow \text{mean}_{|\text{T}(E_{vis}^*)|} W(\text{T}(E_{vis}^*)) \cdot \text{CE}(\text{T}(E_{vis}^*)|1)$
{Calculate pseudo-label loss with soft labels and cross-entropy}

$\text{T}(E_{vis}^*) \leftarrow \sigma(\gamma \cdot \max_{|Att|} \text{Sim}(\text{M}(E_{vis}^*), E_{att}) + \beta)$
{Compute VAP (pseudo-label visual embeddings) using sigmoid and MLP}

Output $\mathcal{L}_{pseudo}$ {Output the final pseudo-label loss}

---

ity of these attributes is then used to set the in-distribution probability, which is combined with the OOD probability to identify unknown objects. However, as noted earlier, in incremental learning, it is challenging to balance the contribution of newly added attributes for new categories. In the official implementation, annotations from previously seen categories are also used in incremental learning, which violates the OWOD task setup. To maintain the integrity of the OWOD setup, we utilize all attributes to predict unknown objects without attribute selection. Theoretically, this modified approach can achieve higher performance metrics.

### G.2. Comparison

**U-Recall** (Table 7). OW-VAP demonstrates a significant performance advantage. On the MOWODB benchmark, OW-VAP surpasses FOMO-ZS+ by 34.8 U-Recall and FOMO-ZS+IN by 6.4 U-Recall. Even when compared to FOMO, which uses all attributes, our method leads by 4.2 U-Recall. This advantage is maintained in subsequent incremental learning phases. On the SOWODB benchmark, our OW-VAP also achieves consistent superiority. In Task 1, we exceed FOMO-ZS+ by 51.5 U-Recall and FOMO-ZS+IN by 15.5 U-Recall. Similarly, against FOMO, as in the MOWODB benchmark, we achieve consistent leadership, with an increase of 4.5 U-Recall in Task 1, 5.1 U-Recall in Task 2, and 4.9 U-Recall in Task 3. Additionally, in the performance for known classes, our method also exhibits significant advantages. In Task 4, we surpass FOMO by 7 mAP and 15.4 mAP, respectively.

**U-AP** (Table 8). Regarding U-AP, our method also demonstrates an advantage. Specifically, in Task 1, OW-VAP surpasses FOMO-ZS+, FOMO-ZS+IN, and FOMO by +6.2, +9.5, and +8.1 U-AP, respectively. In Tasks 2 and 3, our method maintains its lead, achieving advantages of 4.4 and 5.7 U-AP over FOMO, respectively. Our approach also excels in WI and A-OSE metrics, consistently outperforming all FOMO series across the three tasks. However, in terms of A-OSE, we slightly trail behind FOMO. The A-OSE metric of our method could be further improved by labeling more low-confidence known predictions as unknown objects. However, to preserve OW-VAP's performance on known classes, we only convert predictions where unknown confidence exceeds known confidence. Despite this, we only slightly lag behind FOMO in A-OSE, with a confusion count of 201.

**Qualitative comparison** (Figure 6). Figure 6 presents a visualization analysis comparing our method with the FOMO series. Compared to FOMO-ZS, our approach demonstrates stronger unknown recall capability. OW-VAP successfully recalls both the kettle and the mug on the table, whereas FOMO only recalls the kettle. In comparison with FOMO-ZS+IN, OW-VAP exhibits higher precision. FOMO-ZS+IN focuses on internal details of people, such as parts of clothing, which are typically not the objects of interest. Conversely, OW-VAP focuses on objects across the entire image, detecting items like the wooden bridge helmet. When

compared to FOMO, our method also shows advantages. FOMO incorrectly classifies a chair as an unknown object and overlooks items on the table. In contrast, our OW-VAP identifies the kettle and bread, and correctly predicts the chair.

## H. Pseudo Code

The Visual Attribute Parser (VAP, Algorithm 1). First, the input attributes (Att) and class names (Name) are encoded using a text encoder, producing corresponding attribute embeddings ($E_{att}$) and class embeddings ($E_c$). Meanwhile, the visual input image ($I$) is passed through a visual encoder to obtain the visual embedding ($E_{vis}$). The algorithm then computes a matching score by minimizing the distance between the class embedding ($E_c$) and the visual embedding ($E_{vis}$). To ensure high-quality pseudo-labeling, three conditions are applied: (1) the similarity between the visual and attribute embeddings should exceed the mean similarity of the visual and positive attribute embeddings, (2) the similarity must exceed a threshold parameter ($\alpha$), and (3) the top $\delta$ similarities are considered. If all conditions are satisfied, a pseudo-label indicator ($S_{vap}$) is assigned, guiding the learning process.

The Probabilistic Soft Label Assignment (PSLA) algorithm addresses optimization conflicts caused by background interference in pseudo-label generation (Algorithm 2). To resolve this, PSLA estimates the probability of a pseudo-label being a true positive by modeling the loss distributions of positive and negative samples. The algorithm assumes that the loss for background samples is higher and more consistent, while the loss for positive samples is sparser and exhibits greater fluctuations. Using a Weibull distribution (Wb) for both positive and negative samples, PSLA applies maximum likelihood estimation to determine distribution parameters, with a moving weighted average applied to reduce fluctuations in the positive sample distribution. The final soft labels are derived by normalizing the positive and negative sample distributions. These soft labels are then used to weight the pseudo-label loss, which is calculated using the cross-entropy between the predicted labels and the target labels. The PSLA framework enhances the robustness of pseudo-labeling by addressing background noise and optimizing the learning process.

