# OpenReview forum: "OW-VAP: Visual Attribute Parsing for Open World Object Detection"
_ICML.cc/2025/Conference — ICML 2025 poster_

### Official Review · Reviewer_B8QV · 2025-03-11

**Overall Recommendation:** 3

**Summary:**

This paper proposes OW-VAP for OWOD, which does not rely on guidance from LLM. It introduces the visual attribute parser (VAP) to parse
the visual attributes corresponding to the current region. The proposed OW-VAP surpasses the state-of-the-art (SOTA) methods with an advantage of over 13 U-Recall and 8 U-AP in unknown detection.

**Claims And Evidence:**

yes.

**Essential References Not Discussed:**

Most related works are cited properly in the paper.

**Experimental Designs Or Analyses:**

yes.

**Methods And Evaluation Criteria:**

yes.

**Other Comments Or Suggestions:**

N/A

**Other Strengths And Weaknesses:**

**Strengths**

- This paper proposes OW-VAP for OWOD which does not rely on guidance from LLM, by introducing the visual attribute parser (VAP) to parse the visual attributes corresponding to the current region.
- The proposed OW-VAP surpasses the state-of-the-art (SOTA) methods with an advantage of over 13 U-Recall and 8 U-AP in unknown detection.


**Weaknesses**
- The idea of visual attribute parser is not novel. It is similar to the previous traditional zero-shot learning paradigm ([A], [B]). These works should be properly cited and discussed in this work.
- This work uses coarse attributes instead of fine-grained attributes for VAP training. What if the training is based on fine-grained attributes? The results should be displayed for discussion. Besides, the reasons why coarse attributes work than fine-grained ones deserve further analysis.
- There are many hyperparameters involved in the paper, making the results less reproducible.



[A] Zero-Shot Learning—A Comprehensive Evaluation of the Good, the Bad and the Ugly

[B] Learning To Detect Unseen Object Classes by Between-Class Attribute Transfer

**Questions For Authors:**

N/A

**Relation To Broader Scientific Literature:**

The work reveals that previous works resly on the attributes provided by Large Language Model (LLM). Consequently, their performance is highly susceptible to the accuracy of attribute descriptions by LLM. Besides, in subsequent incremental learning, it is challenging
to quantify the contribution of each attribute to the unknown class, leading to difficulties in attribute selection.
Thus, this paper proposes a new detection framework.

**Theoretical Claims:**

This paper doesn't involve much proof of theory.

---

> ### Author Rebuttal · Authors · 2025-03-27
>
> Thank you very much for your valuable comments and suggestions on our manuscript. Below, we provide detailed responses to your feedback:
>
> **Q1**: Similarity to Prior Work
>
> While the referenced works also utilize textual attributes, our OW-VAP differs significantly from these methods in several key aspects:
>
> 1. **Different Task Domain**: The referenced works focus on open-set classification tasks, aiming to train a classifier capable of handling new classes. In contrast, our method addresses the open-world object detection (OWOD) task, which stems from object detection. Specifically, OWOD consists of two parts: novel class discovery and incremental learning. The former requires the detector to identify objects not seen in the training data, while the latter involves fine-tuning the detector to adapt to new tasks based on existing knowledge. For details on OWOD task settings, please refer to  $\textcolor{red}{\text{Appendix B}}$.
>
> 2. **Different Data Requirements**: The referenced methods assume datasets with explicit attribute annotations for training, which is costly to obtain. In contrast, our approach does $\textcolor{red}{\text{not require any additional manual annotation efforts}}$. For example, the MOWODB benchmark is a combination of the VOC and COCO datasets, where the COCO dataset alone contains $\textcolor{red}{\text{over 1.5 million training instances}}$. It is practically impossible to annotate every object in such a large dataset. Instead, VAP training is seamlessly integrated into the detector training process.
>
> 3. **Different Methodology**: The referenced approaches employ statistical techniques such as Bayesian inference for unknown class prediction. In contrast, our method leverages deep learning to map visual embeddings of known classes into a general attribute space. In essence, our VAP functions more similarly to a clustering method.
>
> **Q2**: Coarse Attributes
>
> Our use of coarse textual attributes provides several advantages:
>
> 1. Coarse attributes utilize the same textual descriptions (only 10 in total) across all objects, making them more robust when transferring to new objects compared to fine-grained attributes , which heavily depend on the accuracy of LLMs. For example, in incremental learning, our OW-VAP can directly transfer previously learned knowledge without requiring additional training. In contrast, methods based on fine-grained attributes require generating new attributes followed by an additional round of selection and training ($\textcolor{red}{\text{Table 7 and 8}}$).
>
> 2. As the number of known classes increases, the number of fine-grained attributes also grows. For example, in Task 4 of the MOWODB benchmark, the number of fine-grained attributes reaches $\textcolor{red}{\text{over 8K}}$. Using such fine-grained attributes would be impractical for OW-VAP because it would significantly reduce real-time performance and hinder application in real-world scenarios.
>
> 3. We compared OW-VAP with fine-grained attribute methods (e.g., FOMO series) in Tables 7 and 8. OW-VAP achieves notable performance advantages, $\textcolor{red}{\text{with 4+ improvements in U-Recall and 8+ improvements in U-AP}}$. For further comparative analysis, please refer to $\textcolor{red}{\text{Appendix G}}$.
>
> For insight into why VAP is effective, please see our response to Reviewer sWQp (Weak 1).
>
> **Q3**: Hyperparameter Reproducibility
>
> Our method involves only $\textcolor{red}{\text{three hyperparameters}}$, whose impact on detector performance is summarized in Table 6. For additional details, please refer to Appendix F. Moreover, the model's performance is not highly sensitive to these hyperparameter settings, meaning they can be shared across applications without modification.
>
> Regarding reproducibility, as detailed $\textcolor{red}{\text{in the Code section of Appendix E}}$, all training logs—including weight files, training logs, and visualized results—will be released following code review for transparency and replication purposes.
>
> We sincerely hope that our clarifications resolve remaining questions and provide further insight into our method. If you have additional feedback or need further assistance, please feel free to contact us.

---

> > ### Comment · Reviewer_B8QV · 2025-04-05
> >
> > Thanks for your response.
> >
> > Though most concerns are addressed, the response of Q1 is less convincing.
> > As far as I know, the idea of OW-VAP is not very relevant to object detection, but rather more to do with object recognition. Besides, mapping visual embeddings of known classes into a attribute space is exactly a common paradigm ([C,D]) widely used in traditional ZSL task. So I think a proper discussion between the overall idea in this work and the previous ZSL task is necessary.
> > I keep current rating.
> >
> > [C] Attribute Prototype Network for Zero-Shot Learning NeurIPS2020
> >
> > [D] Goal-Oriented Gaze Estimation for Zero-Shot Learning CVPR2021

---

> > > ### Author Response · Authors · 2025-04-07
> > >
> > > Thank you for your thoughtful comments. We appreciate the opportunity to address your concerns, especially regarding the relationship between OW-VAP and ZSL, and its contributions to OWOD. Our answers to your questions about innovation are as follows:
> > >
> > > __1.__ Regarding whether our contribution is limited due to relying on attribute-based prediction of unknown classes
> > >
> > >   As demonstrated in [B], they employ the attributes of known classes to predict unknown classes, and [C] and [D] continue to follow this paradigm. Nevertheless, [C] and [D] address key technical bottlenecks within the zero-shot learning (ZSL) task, achieving significant breakthroughs that have been widely recognized by the research community. Similarly, [1] introduced the [2] into the field of object detection, establishing a novel paradigm and significantly advancing this domain.  Therefore, we believe that __evaluating the significance of innovation should focus not on whether a theoretical paradigm is extended, but on whether it advances the field or addresses challenges in existing methods__.   For example, [1] tackles the post-processing problem of non-maximum suppression (NMS) in dense predictions, which is a non-parallelizable algorithm that significantly reduces the inference speed of detectors. Similarly, we argue that OW-VAP has contributed to the advancement of open-world object detection (OWOD).  Its advantages and the problems it addresses compared with previous detectors are as follows:
> > >
> > > - Compared to [B], [C], and [D], OW-VAP does not require additional annotation of detailed attributes for known instances in the training data. This indicates that __OW-VAP can extend object detection tasks from closed-set detection to open-world scenarios without the need to separately annotate extra detailed attributes for known classes, thereby avoiding additional annotation costs__.
> > >
> > > - Compared to the FOMO series methods, which rely on large language models (LLMs) to generate detailed class attributes, OW-VAP is more robust. It uses only a small amount of coarse-grained attribute information to predict unknown classes, avoiding the impact of attribute selection and the accuracy of LLM-generated attributes on performance. This robustness is particularly evident in incremental learning scenarios, where OW-VAP demonstrates greater flexibility (see Appendix G for more details).
> > >
> > > - In comparison with prior OWOD methods, OW-VAP requires training only a lightweight visual attribute parser (VAP) to achieve significant improvements in unknown detection performance. Specifically, it surpasses OVOW and FOMO (both of which use the same OVD detector) by a notable __+8 U-AP__. For a comprehensive comparison of these OWOD detectors, please refer to Figure 1.
> > >
> > > Based on these facts, we argue that OW-VAP can serve as a new benchmark, offering fresh research perspectives for future OWOD studies.
> > >
> > > __2__. Our OW-VAP innovation encompasses multiple aspects
> > >
> > > In addition to leveraging coarse-grained attributes to predict unknown classes, __OW-VAP proposes key components during the training of the (VAP), including background sample mining and probabilistic label soft assignment (PLSA)__. As shown in Figures 5 and 6, OW-VAP must be capable of simultaneously locating all unlabeled objects of interest in a given image. Without background sample mining and PLSA, OW-VAP would be limited to focusing only on known classes, resulting in a drop in recall rate. For more details, please refer to the ablation study section of our paper.
> > >
> > > __3__. Differences between OWOD and ZSL
> > >
> > > OWOD and ZSL differ fundamentally in their task objectives and technical challenges. ZSL focuses on classification tasks where the goal is to generalize directly from known classes to unknown classes after training (e.g., [C][D]), without requiring incremental learning. In contrast, __OWOD aims to identify, locate, and subsequently support incremental learning for unknown instances (i.e., the model must dynamically expand its knowledge base)__.
> > >
> > > OWOD also addresses detection-specific challenges, such as object localization issues caused by background interference and densely distributed objects, which are not typically encountered in ZSL, as ZSL primarily focuses on classification. Furthermore, OWOD requires the model to avoid catastrophic forgetting during incremental learning, whereas ZSL is usually designed for one-time learning.
> > >
> > > __For a more detailed introduction to the OWOD task, please refer to Appendix B__.
> > >
> > > The above constitutes our responses to your comments. Furthermore, In the final version, we will add a section to discuss the similarities and differences between OW-VAP's attribute prediction approach and ZSL. We sincerely appreciate your insightful comments and constructive suggestions for improving our work. If you have any further feedback, please feel free to contact us at any time.
> > >
> > > [1] End-to-End Object Detection with Transformers, ECCV2020
> > >
> > > [2] Attention Is All You Need, NeurIPS2017

---

### Official Review · Reviewer_sWQp · 2025-03-13

**Overall Recommendation:** 4

**Summary:**

The paper proposes OW-VAP, a novel framework for Open World Object Detection (OWOD) that eliminates reliance on Large Language Models (LLMs) for attribute descriptions. OW-VAP employs a Visual Attribute Parser (VAP) to extract generic attributes (e.g., shape, color) from visual regions and uses Probabilistic Soft Label Assignment (PSLA) to mitigate optimization conflicts caused by noisy pseudo-labels. Evaluated on MOWODB and SOWODB benchmarks, OW-VAP achieves significant improvements over state-of-the-art methods, surpassing the generalization upper bound of Open Vocabulary Detection (OVD) in Task 2 of MOWODB with a 56.3 U-Recall.

**Claims And Evidence:**

Yes, the claims made in the submission supported by clear and convincing evidence.

**Essential References Not Discussed:**

No.

**Experimental Designs Or Analyses:**

The experiments are well-designed, with comprehensive comparisons against multiple baseline models. Strengths include:
1、performance evaluation on OWOD benchmarks with clear improvements,
2、ablation studies demonstrating the contribution of each component.

**Methods And Evaluation Criteria:**

Yes, the evaluation metrics and dataset used in this paper are suitable for the current scenario.

**Other Comments Or Suggestions:**

No.

**Other Strengths And Weaknesses:**

Strengths:
1、OW-VAP eliminates reliance on LLMs, reducing dependency on text-based attributes.
2、Compared with the SOTA OWOD method, there has been a significant improvement in performance.
3、This paper conducts a thorough experimental validation on MOWODB/SOWODB.

Weaknesses:
1、The attribute learning process lacks deeper theoretical analysis, and further insights on why VAP generalizes well to unknown classes would be helpful.
2、The theoretical basis for PSLA probability modeling is limited.
3、Figure 3 cannot clearly demonstrate the training and inference process of the model. Many texts and graphics are not explained in captions or figures, such as, VLFPN, Weibull, circles in Unknown Object Mining, etc.

**Questions For Authors:**

1、How the attribute descriptions in Table 4 were proposed?
2、The choice of Weibull distribution for modeling loss and the heuristic weighting in Equation (7) lack theoretical justification. Can the author provide ablation studies to explore alternative distributions (such as Gaussian distributions) or validate parameter sensitivity?

**Relation To Broader Scientific Literature:**

The paper effectively positions itself within OWOD research and compares against ORE, OW-DETR, SGROD, KTCN, SKDF, and other recent approaches. The use of YOLO-World as a base model aligns well with modern OVD/OWOD trends.

**Theoretical Claims:**

I check the correctness of proofs for theoretical claims.
The theoretical contributions of the paper are well-motivated. The introduction of VAP and its role in parsing object attributes is sound. However, while PSLA effectively assigns soft labels to mitigate optimization conflicts, a more in-depth theoretical analysis of its probabilistic modeling would strengthen the claim. Specifically, the use of the Weibull distribution for loss distribution estimation warrants further justification.

---

> ### Author Rebuttal · Authors · 2025-03-26
>
> We sincerely appreciate you taking the time to carefully review our manuscript. Your expert opinions and constructive suggestions are invaluable to our research. Below, we provide our responses to the concerns you raised:
>
> **Weak  1**: Why is VAP effective?
>
> Our OW-VAP is built upon the standard YOLO-World, which is a real-time open-vocabulary detector (OVD). YOLO-World is capable of open-vocabulary visual detection by leveraging training on large-scale datasets to align visual embeddings with textual attributes. This alignment not only supports known categories but also provides, in theory, a generalization capability for unknown categories. This is because the relationship between visual embeddings and textual attributes can be understood as a shared mapping in a latent space.
>
> Specifically, OW-VAP utilizes visual embeddings generated by the trained OVD detector and designs a transformation mechanism to align these embeddings with predefined coarse attributes. This transformation mechanism acts as a mapping function from the visual embedding space to the attribute space. Leveraging the shared characteristics of this mapping function, OW-VAP is able to learn common cross-category representations from known categories, enabling semantic generalization to unknown categories. The theoretical basis for this generalization can be attributed to the following points:
>
> 1. The OVD detector gains strong visual-text alignment knowledge through pretraining on large-scale datasets.
>
> 2. The attribute space is intentionally designed to be coarse and abstract, reducing significant differences among object categories and thereby enhancing the generality of intermediate representations.
> 3. The transformation mechanism between visual embeddings and attributes prioritizes shared information instead of specific category features.
>
> Additionally, our method enhances OW-VAP's capability to detect unknown categories by utilizing the pretrained knowledge of the OVD detector. This pretrained knowledge not only supports effective description of known categories but also establishes a robust foundation for aligning visual inputs with textual attributes for unknown categories. For instance, as demonstrated in Table 3, OW-VAP achieves a U-Recall of 45.6 and a U-AP of 10.4 by only using a combination of coarse textual attributes with out-of-distribution probabilities.
>
> **Weak 2**: The annotations in Figure 3
>
> In Figure 3, we illustrate the training process of VAP. In the figure, the blue circles represent samples already labeled in the training set. During training, these samples are treated as positive instances, and we train VAP to pull them closer to their nearest coarse descriptions (represented as yellow circles in the figure). The triangles, on the other hand, represent potentially unlabeled objects within the dataset. These unlabeled objects are further differentiated using the three selection rules outlined in Section 3.2. This process identifies potential objects from unlabeled samples (which correspond to background samples in a closed-set training setting). As such, we refer to this process as "background object mining." Additionally, you may refer to our explanation of this procedure in response to @Drok for further details.
>
> **Q1**: Coarse Attributes
>
> FOMO utilizes large language models (LLMs) to generate detailed attribute descriptions. Before creating the detailed descriptions, it categorizes the object attributes into 10 groups and then generates descriptions within each group. In our approach, we focus only on coarse descriptions. To achieve this, we use the 10 attribute categories and expand them to generate sentences indicating whether each attribute $\textcolor{red}{\text{is significant or not}}$ (as shown in Table 4).
>
> **Q2**: Choice of the Base Probability Model
>
> In fact, $\textcolor{red}{\text{this is a default choice in the OWOD field}}$. Since ORE introduced Weibull modeling for energy-based scenarios, the Weibull distribution has become the standard for similar modeling tasks (e.g., as used in MEPU). Of course, other distributions, such as the Beta distribution, could also be explored and used as a tunable hyperparameter to achieve better performance. However, introducing alternative distributions would increase the complexity of the method, which could hinder further research by the community. Therefore, we adopt the default setting in the OWOD field and use the Weibull distribution to demonstrate the effectiveness of PLSA. Moreover, using a Gaussian distribution is not feasible, as the shape of the distribution is not symmetric and does not match the characteristics of the data.
>
> We hope these clarifications adequately address your concerns regarding our methodology. Should any questions remain, please feel free to contact us for further discussion. We sincerely appreciate your thorough review and valuable feedback.

---

### Official Review · Reviewer_Drok · 2025-03-14

**Overall Recommendation:** 3

**Summary:**

This paper proposes a novel OWOD framework, termed OW-VAP, which operates independently of LLM and requires only minimal object descriptions to detect unknown objects.

**Claims And Evidence:**

Yes, the author provides extensive experimental results to demonstrate the claims.

**Essential References Not Discussed:**

The author has discussed most of related works.

**Experimental Designs Or Analyses:**

Yes. For issues about the experiments, please refer to the below part.

**Methods And Evaluation Criteria:**

Yes.

**Other Comments Or Suggestions:**

N/A

**Other Strengths And Weaknesses:**

Weakness:
1. I think the core idea of this method is using an LLM as the attribute parser, enabling unknown object detection by considering a more comprehensive set of attributes. However, I think this core idea is entirely unrelated to the problem of open-world detection. The authors are merely leveraging an LLM to provide additional information, which I find kind of trivial. I think the author should provide more motivation and justification.
2. Although experimental results show improvement over existing methods, I believe the comparison is kind of unfair. The use of an LLM inherently increases inference time, making the method slower. The author should provide comparision about the model efficiency.

**Questions For Authors:**

N/A

**Relation To Broader Scientific Literature:**

This paper perhaps can be a good contribution to the literature.

**Theoretical Claims:**

There is no theoretical claims in this paper.

---

> ### Author Rebuttal · Authors · 2025-03-26
>
> We sincerely appreciate you taking the time to carefully review our manuscript. Your professional insights and constructive suggestions are invaluable to our research work. Before addressing the questions, we present a complete description of the overall OW-VAP process.
>
> **All Pipeline**
>
> OW-VAP adopts YOLO-World as its base model. YOLO-World is a real-time open-vocabulary detector. During inference, compared to YOLO-World, we additionally introduce a Visual Attribute Parser (VAP), which consists of only $\textcolor{red}{\text{a lightweight multi-layer perceptron (MLP) (e.g., with three layers)}}$.
>
> Given an image $I$ a list of known class names $Name$, and coarse textual attributes $Att$, these inputs are fed into text and visual encoders to produce visual and textual embeddings: $E_{vis} \in \mathbb{R}^{w \times h \times d}, E_{c} \in \mathbb{R} ^ {c \times d} $ and $E_{att} \in \mathbb{R} ^ {n \times d}$. Here, $w$  and $h$, $c$, and $n$ represent the width and height of the feature map (downsampled compared to the original image), the number of known class names, and the number of coarse textual attributes, respectively. It is important to note that $E_{vis}$ is distributed across three feature levels, namely {C3, C4, C5}. For simplicity, we only provide the shape at one feature level. Subsequently, the visual $E_{vis}$ and known class names embeddings $E_{c}$ are passed into the VLPAN to enable interaction between visual and textual features.
> \begin{equation}
>  X' _ {l} = X _ {l} \cdot \sigma(\underset{j \in \{1, ..., c\}}{max}(X _ {l}E _ {c}^\top))^{\top}.
> \end{equation}
> Here, $X_{l}$ represents the visual embeddings at different feature levels {C3, C4, C5}, and $\sigma$ denotes the Sigmoid function.  The resulting output $X_{l}'$ retains the same shape as the original inputs $X_{l}$. Each feature level is divided into a 3 $\times$ 3 grid, and max-pooling is performed within each grid cell. Across all levels, a total of 27 mbeddings are generated as $\widetilde{X}$. Finally, multi-head attention is applied to weight the generated embeddings and the original inputs:
> $$ W' =  E_{c} + \text{MultiHead-Attention}(E_{c}, \widetilde{X}, \widetilde{X}) $$. The final weighted result retains the same shape as the original inputs $E_{c}$. Then, $W'$ will be used to replace $E_c$. In summary, VLFPN first enables interaction between visual and textual embeddings and then performs further fusion using multi-head attention to update $E_c$.
>
> Then, the visual embeddings, along with the updated textual embeddings, are used to decode bounding boxes and generate confidence scores for known classes using cosine similarity. Our Visual Attribute Praser (VAP) operates on the visual embeddings, aiming to extract predefined textual attributes from them. In other words, VAP maps the visual embeddings into the attribute textual space. For details on training VAP, please refer to Section 3 and Appendix E. Afterward, as described in Section 3.4, we use in-distribution and out-of-distribution probabilities to assign confidence scores to unknown classes. An overview of the entire process is briefly described in Section 3.1.
>
> **Q1**: On the use of LLMs
>
> As described in the previous pipeline, $ \textcolor{red}{\text{OW-VAP does not utilize any LLMs during the inference stage}}$. It is built upon the standard, real-time open-world object detection (OWD) model, YOLO-World, with only the addition of a lightweight multi-layer perceptron (MLP). In fact, for object detection tasks, the use of LLMs during inference is generally discouraged. This is because the goal of the community is to achieve a real-time, high-accuracy detection model. Thank you again for evaluating the practical applicability of OW-VAP; we hope these clarifications address your concerns.
>
> **Q2**: On Comparisons
>
> $\textcolor{red}{\text{In the experiments section and appendix}}$, we provide comparisons between OW-VAP and OWOD detectors utilizing foundation models. These include MEPU-FS based on Free-SOSO, SGROD, SKDF, and KTCN based on SAM, as well as ovow and FOMO, which are built on the $\textcolor{red}{\text{same OVD detector}}$ as ours. Overall, OW-VAP achieves comprehensive performance advantages. For example, in Task 1 of MOWODB, OW-VAP outperforms ovow by +13 in U-Recall and +8 in U-AP. For a detailed comparison between OW-VAP and these foundation model-based methods, please refer to Fig. 1.
>
> We hope these responses resolve concerns regarding the methods we employed. Should you have any further questions, please feel free to reach out to us.

---

> > ### Comment · Reviewer_Drok · 2025-04-03
> >
> > The author rebuttal has addressed my concerns, and I will keep my original weak accept rating.

---

> > > ### Author Response · Authors · 2025-04-07
> > >
> > > We would like to sincerely thank you once again for your dedicated efforts during the review process. If you have any further questions or requests, please do not hesitate to contact us.

---

### Official Review · Reviewer_oH8t · 2025-03-20

**Overall Recommendation:** 3

**Summary:**

This paper tackles open world object detection in a fresh way, moving beyond heavy reliance on language models. Their key idea is a Visual Attribute Parser, learning directly from images, combined with smart pseudo-labeling using probabilistic soft labels. Experiments show their approach really pushes the state-of-the-art, hitting impressive recall and precision for unknown objects.

**Claims And Evidence:**

Yes, the claims made in this submission are, for the most part, convincingly supported by the evidence provided. The authors present comprehensive benchmarking against a strong set of baselines on standard OWOD datasets (MOWODB and SOWODB). Ablation studies effectively isolate the contributions of both the Visual Attribute Parser and the Probabilistic Soft Label Assignment, further bolstering the claims regarding the effectiveness of these proposed components. Qualitative visualizations also provide intuitive support for the method's improved detection of unknown objects.

**Essential References Not Discussed:**

The paper does a reasonable job citing relevant OWOD methods. However, to fully contextualize OW-VAP's contribution, briefly mentioning foundational works in Open Set Recognition (OSR), like the early work of Scheirer et al. on zero-shot learning and open set recognition, would be beneficial.

**Experimental Designs Or Analyses:**

The experiments are strong overall, especially the benchmark comparisons and ablations. However, it's not entirely clear why the vague attribute descriptions were chosen over more specific ones. It would strengthen the paper to see some analysis or even a small experiment exploring the impact of using different levels of attribute description granularity.

**Methods And Evaluation Criteria:**

Yes, the proposed methods and evaluation criteria are well-suited and sensible for the problem of Open World Object Detection. The introduction of the Visual Attribute Parser and Probabilistic Soft Label Assignment are conceptually sound and directly address the challenges of detecting unknown objects. The evaluation on MOWODB and SOWODB benchmarks, using U-Recall, U-AP, WI, and A-OSE metrics, aligns perfectly with established practices in the OWOD field and provides a comprehensive assessment of the method's capabilities. The comparison to relevant state-of-the-art methods and thorough ablation studies further strengthen the validity of the evaluation.

**Other Comments Or Suggestions:**

For Figure 3's clarity, explicitly label VLFPN input/output flows and its role in the caption, and also explain the 'Unknown Object Mining' triangle symbol for better reader understanding.

**Other Strengths And Weaknesses:**

Strengths:
1)The OW-VAP framework, especially the Visual Attribute Parser and Probabilistic Soft Label Assignment, presents a genuinely novel and well-motivated approach to OWOD, moving away from overly LLM-dependent methods.
2)The experimental results clearly demonstrate substantial performance improvements over existing SOTA methods on standard benchmarks, which is a significant contribution to the field. Approaching the OVD generalization limit is also noteworthy.
3)The paper is generally well-written and clearly explains the proposed method.

Weaknesses:
1)While the "vague" attribute descriptions are motivated, direct experimental comparison justifying this choice over more specific descriptions is lacking.
2)PSLA hinges on modeling loss distributions with Weibull distributions, but the paper lacks strong evidence why Weibull is the right choice or any analysis of how sensitive PSLA is to this assumption. Justification or robustness checks are needed.

**Questions For Authors:**

Q1: In Table 1, the U-Recall of OW-VAP in Task 2 surpass the Zero-Shot performance, which is counterintuitive. Is the claim that OW-VAP approaches or even exceeds the generalization upper limit of OVD detectors sufficiently justified? Further clarification and stronger evidence are needed.
Q2: You mention using 'vague' attribute descriptions, which makes sense, but have you considered comparing this approach experimentally to using more specific descriptions? It would be interesting to see some evidence justifying this particular choice.
Q3: The PSLA method relies on modeling loss distributions with Weibull distributions. Could you elaborate on why Weibull was chosen and what makes it a good fit here? Also, it would be helpful to see some analysis on how sensitive PSLA is to this assumption, maybe some robustness checks?
Q4: Figure 3 is helpful, but to make it even clearer, could you explicitly label the input and output flows of the VLFPN, maybe in the caption? And could you also explain what the triangle symbol for 'Unknown Object Mining' represents?

**Relation To Broader Scientific Literature:**

OW-VAP directly tackles limitations of LLM-reliant OWOD methods like KTCN and FOMO. The Visual Attribute Parser builds on visual attribute recognition ideas, but applies them uniquely to OWOD, differing from attribute use in closed-set OD or OVD. Probabilistic Soft Label Assignment smartly adapts pseudo-labeling strategies from works like PROB and ORE, specifically for background noise, going beyond simple thresholds.

**Theoretical Claims:**

I didn't spot any formal proofs to really "check" in this submission. It's more of an empirically-driven approach, focused on the algorithm and results. That being said, the conceptual ideas behind VAP and PSLA seem pretty solid and logically motivated for the problem they're tackling.

---

> ### Author Rebuttal · Authors · 2025-03-25
>
> We greatly appreciate your careful and thoughtful review of our manuscript. Your meticulous approach to the review process contributes significantly to the advancement of the research community. Below, we provide our responses to the concerns raised in your comments:
>
> $\textbf{Q1}$: Regarding the question on how the model's performance could reach or even surpass the theoretical upper bound in zero-shot generalization
>
> In object detection tasks, performance evaluation must consider both recall capability (i.e., the ratio of correctly detected targets to total targets) and precision (i.e., the proportion of correct detections among all positive predictions). The mean Average Precision (mAP) metric has become the standard for balancing these competing objectives, as comprehensively explained in [1].
>
> However, conventional mAP formulations predominantly focus on closed-set detection tasks. In the context of open-world object detection (OWOD), certain object categories may lack annotations in the training data, $\textcolor{red}{\text{including the class names}}$. As a result, object detectors typically demonstrate degraded performance in such settings. Under such conditions, direct application of AP metrics becomes inadequate to reflect the performance, as demonstrated by leading OWOD works including ORE, OW-DETR, and PROB. For example, KTCN and PROB achieve similar U-AP values, but in terms of U-Recall, KTCN achieves more than twice the performance of PROB.
>
> Thus, benchmarks such as MOWODB and SOWODB prioritize U-Recall as a metric to evaluate unknown object detection performance. $\textcolor{red}{\text{We surpass the zero-shot performance of the detector only in U-Recall, but not in U-AP}}$. Notably, exceeding zero-shot testing performance in U-AP remains challenging without utilizing additional information about unknown classes. The relative comparison of all methods in U-AP and U-Recall is shown in $\textcolor{red}{\text{Figure 1}}$.
>
> $\textbf{Q2}$: Comparison Between Vague Descriptions and Detailed Descriptions
>
> As mentioned in the paper, detailed descriptions rely on large language models (LLMs) to describe certain aspects of known classes, with the expectation that unknown classes share similarities in these aspects (as demonstrated by FOMO). These similarities are then used to recall unknown objects. However, directly using detailed descriptions in our method is impractical because their quantity exceeds 8,000. In contrast, our method achieves detection of unknown classes using only coarse descriptions composed of 10 simple sentences. Furthermore, we compared our method with all FOMO variants (using detailed attributes) in $\textcolor{red}{\text{Tables 7 and 8}}$. Our method still demonstrated significant advantages, achieving improvements of +4 in U-Recall and +8.1 in U-AP. For more details, please refer to $\textcolor{red}{\text{Appendix G}}$.
>
> $\textbf{Q3}$: Choice of the Base Probability Model
>
> In line with OWOD conventions, we adopt the Weibull distribution for probability modeling, as introduced by ORE for energy-based methods. While exploring alternative distributions (e.g., Beta) could potentially enhance performance, such designs often increase methodological complexity, hindering replicability and adoption. By adhering to the established Weibull distribution, $\textcolor{red}{\text{we ensure both simplicity and effectiveness}}$. Therefore, we follow the default setting in the OWOD field by using the Weibull distribution for modeling, demonstrating the effectiveness of PLSA.
>
> $\textbf{Q4}$: Explanation of Figure 3 Annotations
>
> In Figure 3, we illustrate the training process of VAP. In the figure, the blue circles represent samples that have already been labeled in the training set. These samples are treated as positive samples during training, and we train VAP by pulling them closer to the nearest coarse description (represented by the yellow circles in the figure). The triangles, on the other hand, represent $\textcolor{red}{\text{potential objects}}$. These are further distinguished using the three selection rules described in Section 3.2. We hope this clarifies the visual annotations in Figure 3.
>
> We sincerely thank the reviewer for the valuable feedback. We hope our responses address the raised concerns, and we remain available to clarify any remaining questions or suggestions.
>
> [1] Measuring Object Detection Models - mAP - What is Mean Average Precision?

---

### Decision · Program_Chairs · 2025-05-01

**Decision:**

Accept (poster)

**Comment:**

The manuscript received ratings of 4, 3, 3, and 3. Reviewers appreciated the proposed method and promising results with comprehensive benchmarking against a strong set of baselines on standard OWOD datasets. The reviewers also raised several questions and the authors provided rebuttal to address them. Post-rebuttal, all reviewers remained positive and generally expressed that their concerns are mostly addressed. Given the manuscript, the rebuttal, and that all reviewers are generally positive, the AC agrees with the reviewers and recommend acceptance. Authors are encouraged to take into account all the suggestions of reviewers and rebuttal when preparing the revised manuscript.